# Comparison of methods and resources for cell-cell communication inference from single-cell RNA-Seq data

Daniel Dimitrov[1], Dénes Türei[1], Martin Garrido-Rodriguez [1], Paul L. Burmedi [1], James S. Nagai[2,3], Charlotte Boys[1], Ricardo O. Ramirez Flores [1], Hyojin Kim[1], Bence Szalai[4], Ivan G. Costa [2,3], Alberto Valdeolivas [5,6], Aurélien Dugourd[1,6] & Julio Saez-Rodriguez [1✉]

The growing availability of single-cell data, especially transcriptomics, has sparked an increased interest in the inference of cell-cell communication. Many computational tools were developed for this purpose. Each of them consists of a resource of intercellular interactions prior knowledge and a method to predict potential cell-cell communication events. Yet the impact of the choice of resource and method on the resulting predictions is largely unknown. To shed light on this, we systematically compare 16 cell-cell communication inference resources and 7 methods, plus the consensus between the methods' predictions. Among the resources, we find few unique interactions, a varying degree of overlap, and an uneven coverage of specific pathways and tissue-enriched proteins. We then examine all possible combinations of methods and resources and show that both strongly influence the predicted intercellular interactions. Finally, we assess the agreement of cell-cell communication methods with spatial colocalisation, cytokine activities, and receptor protein abundance and find that predictions are generally coherent with those data modalities. To facilitate the use of the methods and resources described in this work, we provide LIANA, a LIgand-receptor ANalysis frAmework as an open-source interface to all the resources and methods.

[1] Heidelberg University, Faculty of Medicine, and Heidelberg University Hospital, Institute for Computational Biomedicine, BioQuant, Heidelberg, Germany. [2] Institute for Computational Genomics, Faculty of Medicine, RWTH Aachen University, Aachen 52074, Germany. [3] Joint Research Center for Computational Biomedicine, RWTH Aachen University Hospital, Aachen, Germany. [4] Faculty of Medicine, Department of Physiology, Semmelweis University, Budapest, Hungary. [5] Roche Pharma Research and Early Development, Pharmaceutical Sciences, Roche Innovation Center Basel, Basel, Switzerland. [6] These authors contributed equally: Alberto Valdeolivas, Aurélien Dugourd. ✉email: pub.saez@uni-heidelberg.de

Single-cell RNA sequencing (scRNA-Seq) data has become a driving force in the analysis of the cellular heterogeneity of tissues. Furthermore, Spatial Transcriptomics has recently emerged as a technology to measure gene expression while preserving the spatial distribution of cells in a sample, thus providing an unprecedented opportunity to decipher tissue architecture[1]. These advancements have in turn led to an increased interest in the development of tools for cell-cell communication (CCC) inference. CCC events are essential for homeostasis, development, and disease, and their estimation is becoming a routine approach in scRNA-seq data analysis[2]. CCC commonly refers to interactions between secreted ligands and plasma membrane receptors. This picture can be broadened to include secreted enzymes, extracellular matrix proteins, transporters, and interactions that require the physical contact between cells, such as cell-cell adhesion proteins and gap junctions[3]. For simplicity, we refer to all of these events involving protein-protein interactions as CCC.

A number of computational tools and resources have emerged that can be further classified as those that predict CCC interactions alone[4–17], and those that additionally estimate intracellular activities related to CCC[18–24]. Here, we focus on the former (Table 1). These CCC tools typically use gene expression information obtained by scRNA-Seq. In general, single cells are clustered by their gene expression profile and cell type identities are assigned to the clusters based on known gene markers. Then, CCC tools can predict intercellular crosstalk between any pair of clusters, one cluster being the source and the other the target of a CCC event. CCC events are thus typically represented as a one-to-one interaction between a transmitter and receiver protein, accordingly expressed by the source and target cell clusters. The information about which transmitter binds to which receiver is extracted from diverse sources of prior knowledge. Roughly, CCC tools then estimate the likelihood of crosstalk based on the expression level of the transmitter and the receiver in the source and target clusters, respectively. Every tool has two major components: a resource of prior knowledge on CCC (interactions), and a method to estimate CCC from the known interactions and the dataset at hand. Most tools have been published as the combination of one resource and one method, but in principle any resource could be combined with any method.

Despite the aforementioned common premises to explore CCC events, each tool uses a different method, such as permutation of cluster labels, regularisations, and scaling, to prioritise interactions according to the input datasets (Table 1). In turn, these different approaches result in diverse scoring systems that are challenging to compare and evaluate. The difficulties are further exacerbated by the lack of an appropriate gold standard to benchmark the performance of CCC methods[2,25]. Nevertheless, different strategies have been used to indirectly evaluate the methods' performance, including a presumed correlation between CCC predictions and spatial adjacency[14,22], recovering the effect of receptor gene knockouts[22], robustness to subsampling[14], agreement with proteomics[12], simulated scRNA-Seq data[9], and the agreement among methods[10,12,14,22].

The available prior knowledge resources, largely composed of ligand-receptor, extracellular matrix, and adhesion interactions, are typically distinct but often show partial overlap[3,26]. Some of these resources also provide additional details for the interactions such as information about subcellular localisation[3,14], classification into signalling pathways and categories[14,27] (Supplementary Table 1). Notably, some resources[3,8,14,27,28] (Supplementary Table 1), and consequently their corresponding methods, focus on protein complexes as the functional units of CCC, which are crucial for the coordination of signalling as different subunit combinations may induce distinct responses[8]. Despite the fact that CCC inference is constrained by the prior knowledge used,

---

**Table 1 Tools included in the framework.**

| Tool/Method | Resource | Methods' scoring systems |
|---|---|---|
| **CellChat#**[14] | CellChatDB | (1) **Probability**—based on the expression of differentially expressed transmitter and receiver genes and their mediators, calculated with the law of mass action<br>(2) **P-values†**—significance identified via permutation of cell cluster labels and recalculating the probabilities for each cell pair and each transmitter-receiver interaction |
| **CellPhoneDBv2#**[8] | CellPhoneDB | (1) **Truncated Mean**—average expression of transmitter and receivers, the minimum expression (by default) of heteromeric complex of subunits<br>(2) **P-values†**—significance identified via permutation of cell cluster labels to determine a null distribution of means for each receiver-transmitter interaction |
| **Connectome**[10] | Ramilowski | (1) **weight_norm**—derived via the product (by default) of the normalised expression of transmitter and receiver genes<br>(2) **weight_scale†**—derived from a function (mean, by default) of the $z$-scores of the transmitter and the receiver, scaled according to cell cluster specificity |
| **Crosstalk scores** | - | (1) **Crosstalk score†**—Cytotalk-inspired[22] crosstalk scores were derived from the expression of transmitters and receivers, weighted by the likelihood of autocrine signalling between the source and target cell types. |
| **logFC Mean** | - | (1) **logFC Mean†**—iTALK-inspired[6] logFC means, derived using the mean of the logged one-versus-all fold change of receiver and transmitter gene expression |
| **NATMI**[11] | ConnectomeDB | (1) **Mean-expression edge weight**—transmitter and receiver gene expression product<br>(2) **Specificity-based edge weight†**—the mean expression of the transmitter and receiver are divided by the sum of the means of the same transmitters/receivers across all cell clusters |
| **SingleCellSignalR#**[12] | LRdb | (1) **LRscore**—a regularised score calculated using the squared expression of the transmitter and receiver (sqTRE) divided by sum of the mean of the count matrix and sqTRE. |
| **Consensus** | - | (1) **Robust Rank Aggregate**[65]—preferentially highly-ranked interactions are obtained from a distribution generated from the interaction rankings of other methods |

Each method considers expression at the cell cluster level, and all of the scoring systems presented here use the expression of transmitters and receiver genes in the source and target cells, respectively. In addition to the seven methods, we included their consensus.
In bold are the names of cell-cell communication inference methods and their scoring functions.
Dagger (†): Explicitly incorporates communicating cell-pair specificity in interaction predictions
Hashtag (#): CellPhoneDB, CellChat, and SingleCellSignalR provide explicit thresholds to control for false positive interaction predictions. In the case of the former two, these are permutation-based p-values, whereas SingleCellSignalR's LRscore has a suggested threshold of 0.5.
Methods that additionally infer intracellular processes, such as NicheNet[19], Cytotalk[22], and SoptSC[20] are not directly comparable but instead provide complementary analyses.

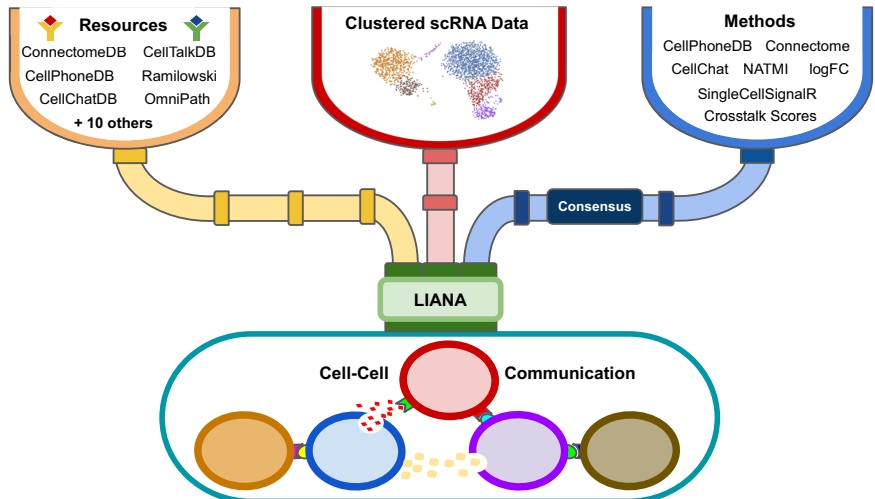

**Fig. 1 LIANA—a LIgand-receptor ANalysis frAmework.** LIANA takes any annotated single-cell RNA (scRNA) dataset as input and establishes a common interface to all the resources and methods in any combination. LIANA also provides a consensus ranking for the method's predictions.

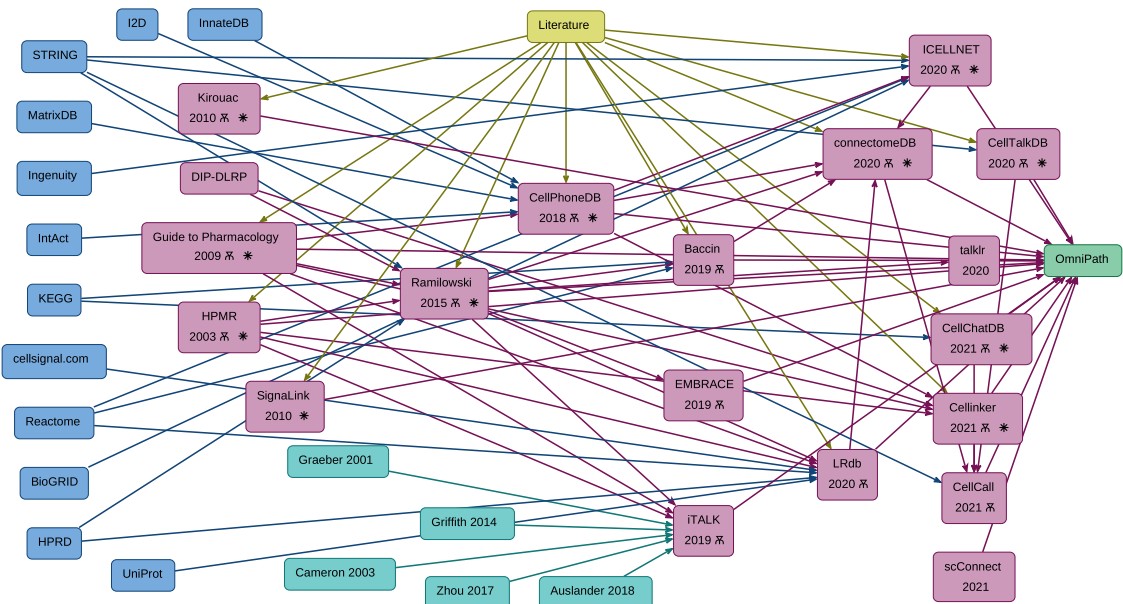

**Fig. 2 Dependencies and overlap between CCC resources.** The lineages of CCC interaction database knowledge. General biological knowledge databases (blue), CCC-dedicated resources (magenta), manual literature curation effort (yellow), additional resources included in iTALK (cyan), and OmniPath (green). Arrows show the data transfers between resources. The yus symbol (ꓭ) indicates the manual-curation of resources, defined by explicitly mentioning that these resources are 'manually' or 'expert' curated. The asterisk ( ∗ ) indicates that the resource was included in the analyses presented here.

yet the impact of resource choice is largely unexplored, with the exception of a descriptive comparison of 4 resources with one method[26]. Thus, it remains unclear how the choice of resource and method affects the results and thereby the biological interpretation of the scRNA-seq data.

In this work, we systematically compared all combinations of 16 resources and 7 CCC methods, plus their consensus (Fig. 1). First, we explored the degree of overlap among resources and whether certain resources are biased toward specific biological terms, such as pathways and tissue-enriched proteins. Then, we analysed how different combinations of resources and methods influence CCC inference by decoupling the methods from their corresponding resources and applying all method-resource combinations on six different datasets. Finally we evaluated the agreement of the different CCC methods with additional

modalities, including spatial adjacency, cytokine activities, and protein abundance. All results were generated using LIANA—a LIgand-receptor ANalysis frAmework (Fig. 1; available at https://github.com/saezlab/liana).

## Results

**Resource uniqueness and overlap**. To investigate the lineages of CCC resources, we manually gathered information about the origins of every resource. Many of these resources share the same original data sources, including general biological databases such as KEGG[29,30], Reactome[31], and STRING[32] (Fig. 2). Moreover, interactions from the Guide to Pharmacology[33], CellPhoneDB[8], HMPR[34], and in particular Ramilowski (FANTOM5)[35], which are manually curated, were commonly incorporated into subsequently published resources (Fig. 2; Supplementary Table 2). All

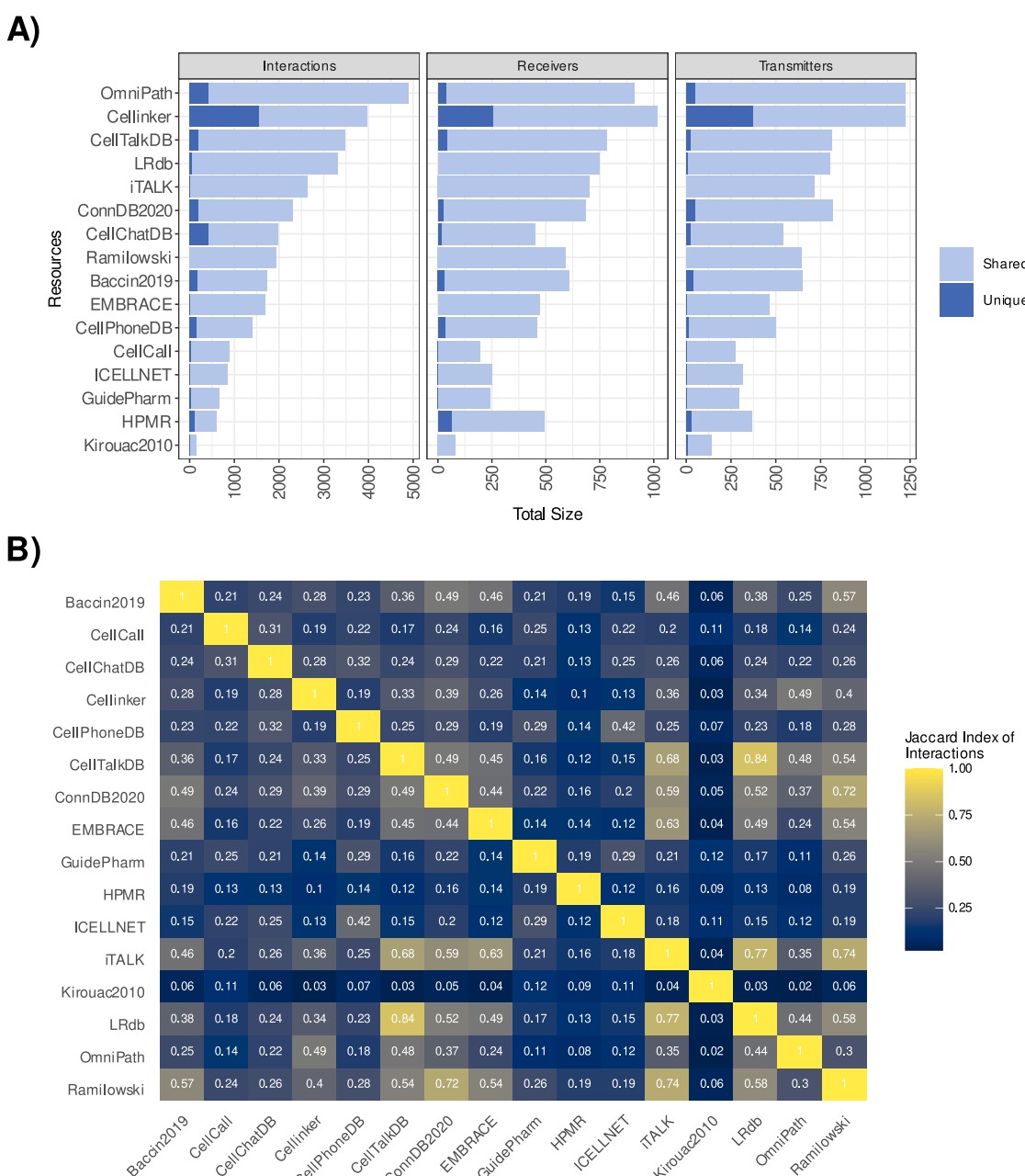

**Fig. 3 Cell-cell communication resources—uniqueness and overlap. A** Shared and unique Interactions, Receivers and Transmitters for each resource. **B** Similarity between the different resources based on the interactions (Jaccard Index). Source data are provided as a Source Data file.

the resources included in this analysis are integrated into OmniPath's CCC resource[3], along with additional CCC interactions from other sources (e.g. SIGNOR[36], Adhesome[37], SignaLink[38]). A part of the OmniPath CCC resource, also referred to as 'OmniPath' and used in this work, was filtered by curation and protein localisation quality ("Processing of CCC resources" Methods).

As a consequence of their common origins, we noted limited uniqueness across the resources, with mean percentages of 6.4% unique receivers, 5.7% unique transmitters, and 10.4% unique interactions (Fig. 3A; Supplementary Table 1). One notable exception was Cellinker's resource[16], as 39.3% of its interactions were not present in any other resource. Despite the fact that few components were unique to any given resource, the pairwise overlap between the resources varied and was often limited (Fig. 3B; Supplementary. Fig. S1). Yet, high similarity was observed between CellTalkDB[26], ConnectomeDB[11], iTALK[6], LRdb[12], and Ramilowski (Fig. 3B). Each of these resources, together with OmniPath and Cellinker, contained an average of at least 60% of the interactions present in other resources, largely explained by each containing a large proportion (>80%) of the interactions present in Ramilowski (Supplementary Fig. S2). Baccin[28], CellPhoneDB, CellChatDB, and EMBRACE showed limited similarity with other resources, as each included on average ~40–50% of the interactions present in any other resource. These latter resources, except EMBRACE, include protein complexes, which were dissociated and treated as distinct protein subunits in our resource analyses. The relatively smaller resources CellCall[23], ICELLNET[13], Guide to Pharmacology, HMPR and Kirouac2010[39] were the most dissimilar from the remainder. Finally, the similarity among the resources was generally higher when considering transmitters and receivers

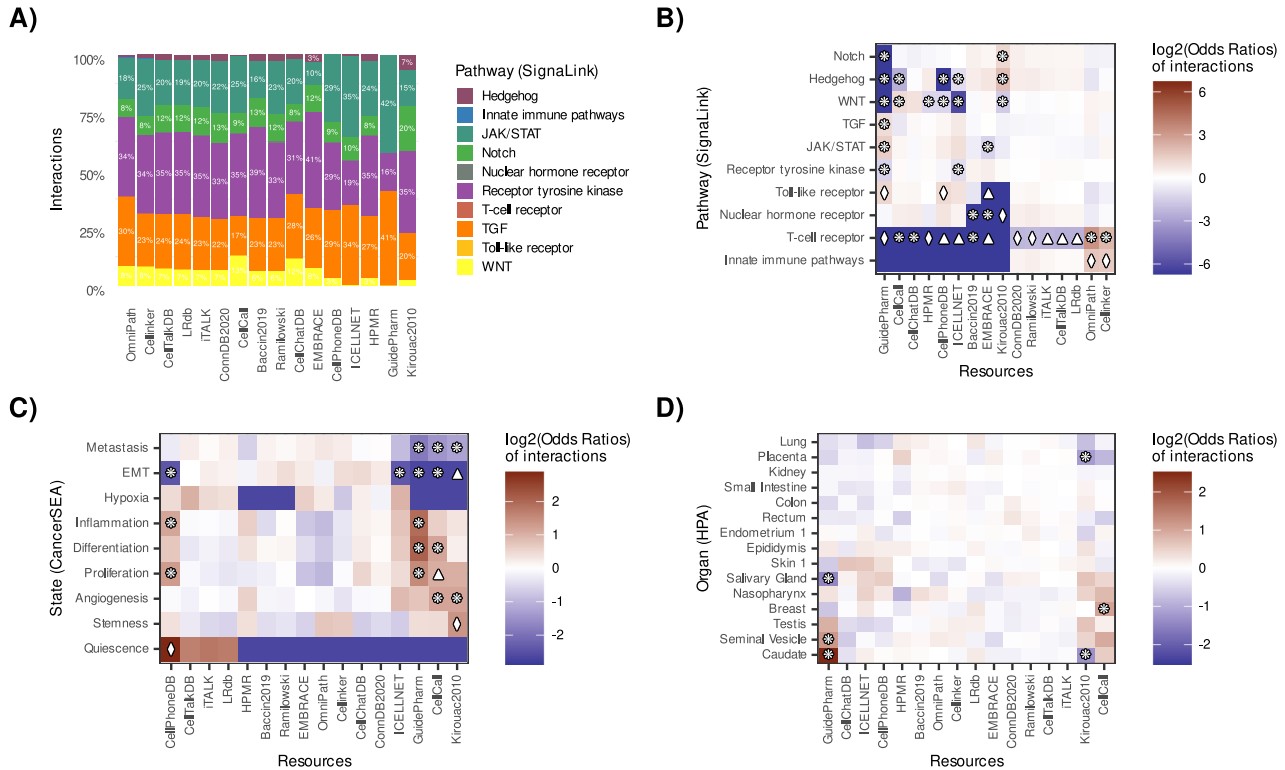

**Fig. 4 Representation of functional categories in CCC resources.** CCC resources distributions in terms of number of interactions (**A**) and relative abundance (**B**) matched to the SignaLink database. Relative abundance of interactions categorised by (**C**) CancerSEA's cancer-related gene sets, and (**D**) organ-enriched proteins from the Human Protein Atlas (HPA). Fisher's exact test was used to estimate the differentially-represented categories. Differentially represented (absolute(log2(Odds ratio)) >1) categories were marked according to FDR-corrected $p$-values =< 0.05 (diamond, ◇), 0.01 (triangle, △), and 0.001 (8-pointed asterisk; ✲). Source data are provided as a Source Data file.

(Supplementary Figs. S1, 2), rather than the interaction themselves, suggesting that different resources account for different interactions between the same proteins.

**Resource prior knowledge bias.** Since CCC inference relies heavily on prior knowledge to estimate intercellular communication events, the choice of resource and any potential bias in it is expected to impact the results. We therefore explored whether the coverage of each CCC resource, when compared to the collection of all resources, is biased toward specific functional categories, tissue-enriched proteins, disease-associated genes, or subcellular locations.

To examine whether specific pathways and biological functions are unevenly represented in CCC resources, we matched the interactions, receivers and transmitters from each resource to well-known pathways and functional categories from SignaLink[38], NetPath[40], and CancerSEA[41] ("Descriptive analysis of resources" Methods) and compared the resulting distributions across 16 CCC-dedicated resources (Supplementary Table 2).

The Receptor tyrosine kinase (RTK), JAK/STAT, TGF, WNT, and Notch pathways covered the largest proportions of interactions matched to SignaLink (Fig. 4A), with analogous results observed for receivers and transmitters (Supplementary Fig. S3). The interactions from Ramilowski, ConnectomeDB, CellTalkDB, LRdb, and iTALK showed a highly similar patterns, explained by the high overlap of these resources, with all of them showing significant underrepresentation of the T cell receptor pathway (Fig. 4B). A more pronounced underrepresentation of the same pathway was observed in Guide to Pharmacology, ICELLNET, CellPhoneDB, CellCall, CellChatDB, HMPR, Baccin2019, EMBRACE, and Kirouac2010. On the contrary, the T-cell

receptor pathway was significantly overrepresented in OmniPath and Cellinker. When we used NetPath instead of SignaLink to define the T-cell receptor pathway, we also observed underrepresentation in HMPR, CellCall, EMBRACE, and Kirouac2010 and overrepresentation in OmniPath (Supplementary Fig. S4A). Moreover, the Signalink WNT pathway was underrepresented in Guide to Pharmacology, ICELLNET CellPhoneDB, HMPR, and Kirouac2010, and on the contrary overrepresented in CellCall. We saw similar results when using NetPath's WNT pathway (Supplementary Fig. S4A). We also observed uneven representations across the resources, in particular for the Hedgehog, Notch, and Innate Immune pathways (Fig. 4A; Supplementary Fig. S4A).

We then matched interactions to cancer-related gene sets from CancerSEA[41], which were also unevenly represented. For example, interactions from the CellPhoneDB resource were overrepresented in gene sets associated with inflammation, proliferation, and quiescence (Fig. 4C; Supplementary Fig. S5). Gene sets associated with epithelial-mesenchymal transition were underrepresented in CellPhoneDB, Guide to Pharmacology, CellCall, ICELLNET, and Kirouac2010. This observation was further supported by the underrepresentation of direct-contact signalling in the latter two resources (See Supplementary Note 1; Supplementary Fig. S6).

We also examined the coverage of tissue-enriched proteins and disease markers from the Human Protein Atlas[42] and DisGeNET[43], respectively. Organ-enriched proteins were largely uniformly distributed across the CCC resources, with some exceptions, such as organ-associated proteins from the Breast, Bone Marrow, Lymph Nodes, and the Hypothalamus (Fig. 4D; Supplementary Figs. S7, S8). Similarly, tissue-enriched proteins were generally distributed evenly across most CCC resources, with some exceptions including the underrepresentation of interactions associated with cardiomyocyte proteins in

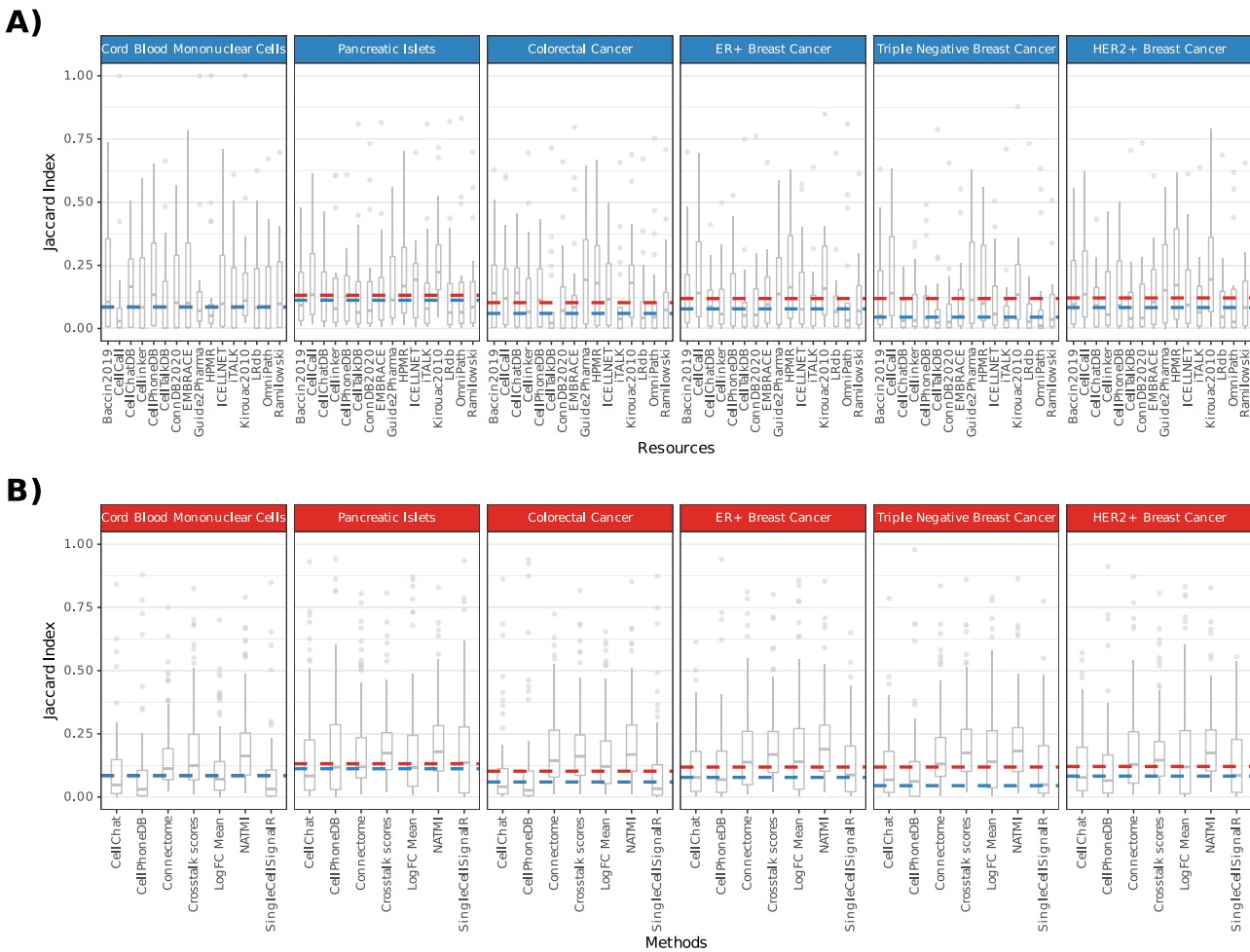

**Fig. 5 Overlap of predictions using any combination of CCC methods and resources.** Overlap (Jaccard index) in the 1000 highest ranked (**A**) when using the same Resource with different Methods (Blue; $n = 7$) and (**B**) when using the same Method with different Resources (Red; $n = 16$). Boxplots represent the median pairwise jaccard index with hinges showing the first and third quartiles and whiskers extending 1.5 above and below the interquartile range. The dashed lines represent the median when using different resources (red) and methods (blue); the lines overlap for the CMBCs dataset. Source data are provided as a Source Data file.

ICELLNET and Kirouac2010, as well as the overrepresentation of proteins associated with Glial cells in Guide2Pharma (Supplementary Figs. S9, S10).

Finally, no differentially represented disease markers were noted in any of the CCC resources (Supplementary Fig. S11).

In summary, our results indicated biases towards certain pathways, functional categories, and tissue-enriched proteins across the different CCC resources, implying that resource choice can influence the functional interpretation of CCC predictions.

**Using LIANA to systematically compare CCC predictions**. To estimate the relative agreement between CCC methods and the importance of the resources, we built LIANA—a framework to decouple tools from their inbuilt resources. LIANA enabled us to combine the 16 CCC resources detailed in the descriptive resource analysis above (Supplementary Table 2), with 7 CCC methods used to prioritise ligand-receptor interactions from scRNA-Seq data (Table 1). We then predicted the interactions from all possible method-resource combinations for 6 single-cell RNA datasets from three different subtypes of breast cancer[44], cord blood mononucleated cells[45], Pancreatic Islets[46], and colorectal cancer[47] (Methods "Data availability").

We first looked at the overlap between the 1000 highest ranked interactions predicted for every method-resource combination.

Whenever available, we used the recommended scoring functions (Supplementary Table 3), each tailored for predicting relevant interactions. We found consistently low overlap in the top predicted interactions when using either different methods or different resources (Fig. 5). The median pairwise Jaccard index when using different methods ranged from 0.045 to 0.112 across datasets (median = 0.080) (Fig. 5A). The overlap when using different resources was slightly higher, as the median pairwise Jaccard index ranged from 0.085 to 0.132 (median = 0.119) (Fig. 5B). We found similar results when considering the top 1% predicted interactions instead of the top 1000 (Supplementary Fig. S12; Supplementary Note 2). These analyses revealed substantial discrepancies in the highest-ranked predicted interactions by the different methods under study.

These discrepancies reflect the diverse nature of the scoring systems used to prioritise interactions of interest, and in particular, the different approaches used to assign communication cell cluster pair specificity to the interactions (marked with a dagger (†) in Table 1; used by all methods except SingleCellSignalR). The low overlap between the results of the different methods was also reflected by dissimilarities in the relative importances assigned to different cell types (See Supplementary Note 2).

On the other hand, the low overlap between the highest ranking interactions using different resources was largely

expected due to the limited overlap between the CCC resources as described in "Resource Uniqueness and Overlap".

Taken together, our results suggest that both the choice of method and the resource can have a considerable impact on the predicted interactions.

**Robustness to noise in resources and data**. We then analysed the sensitivity of the methods to the addition of noise in the data and resource ("Robustness analyses" Methods). We found that most were fairly robust to subsampling of the total number of cells (Supplementary Fig. S13A), while erroneous annotation of cell types had a stronger effect, highlighting the importance of preprocessing and proper cluster annotation (Supplementary Fig. S13B). The methods were also adequately robust to the selective replacement of original canonical resources interactions with spurious putatively false interactions ("Robustness analyses" Methods), in which highest ranked interactions for each method were preserved (Supplementary Fig. S13C). The non-selective replacement of interactions, meant to simulate the change of resource (Supplementary Fig. S13D), had a strong effect on all methods, reflecting the low overlap when using different resources observed in the overlap analysis above.

Overall, our analysis showed that all methods, especially CellChat, CellPhoneDB, and SingleCellSignalR, were fairly robust to noise in both the data and the resource.

**Association between CCC predictions and cytokine expression signatures**. Next, given the lack of a ground truth, we used other data modalities to indirectly evaluate the methods using Omni-Path, the resource with the largest coverage.

First, we noted that all methods appropriately detect specifically-expressed receptor proteins across seven CITE-seq datasets (See Supplementary Note 3). Since protein levels of receptors do not necessarily imply activity, we evaluated the methods' agreement with predicted cytokine activities using 43 cytokine expression signatures[48] on two datasets coming from two subtypes of breast cancer[44] (Methods "Agreement with cytokine signatures"). To show the association between CCC predictions and cytokine activities, we calculated the odds ratios between preferentially ranked interactions and positively enriched cytokines across a range of ranks. We found generally positive trends between cytokine activities and the most prioritised CCC interactions across all methods. The observed trends largely converged toward the random baseline as the number of considered interactions increased (Fig. 6A). Connectome, the Crosstalk scores, and NATMI showed a consistent trend across both datasets, while SingleCellSignalR, logFC Mean, CellChat, CellPhoneDB, and the consensus of the methods (Table 1) showed negative or lack of signal for the higher ranks of the HER2 + dataset (Fig. 6A; Supplementary Fig. S14). Notably, a high agreement with cytokine activities was observed for CellChat and CellPhoneDB in the HER2 + dataset, when considering all of their predictions subsequent to false-positive filtering (vertical line in Fig. 6A), highlighting the value of the false-positive control steps of these methods.

These results suggest that the interactions identified as relevant by all methods were largely concordant with cytokine activities, confirming the agreement of predicted CCC interactions with downstream signalling events.

**Enrichment of predicted interactions between spatially adjacent cell types**. Next, we leveraged spatial information as a way to support the methods' predictive potential, under the assumption that, while many other factors are involved, colocalized cell populations are expected to have a higher chance to interact with each other than other non-adjacent cell types[14,22,49,50]. That is, the highest ranked interactions predicted between various cell populations are expected to be positively associated in interactions between pairs of adjacent cell types (Methods "Agreement with spatially adjacent cell types").

We used the spatial mapping information from eight 10× Visium slides (see Methods), corresponding to a murine brain cortex[51] and triple negative breast cancer[44] datasets, to identify the colocalized cell types in the tissues. We observed a positive trend of increased colocalisation of cell types in Visium and prioritisation of CCC interactions in the scRNA datasets (Fig. 6B). This trend was particularly consistent for the well-structured, murine brain cortex dataset, where all methods, except the Crosstalk scores, showed an association between cell type spatial adjacency and CCC predictions, with Connectome, LogFC Mean, and the consensus displaying the most positive associations. In the case of the triple negative breast cancer dataset, only the predictions by the consensus and LogFC Mean showed a consistent, positive association with spatial adjacency (Fig. 6B).

We conducted a similar analysis with seqFISH[52] and merFISH[53] datasets ("Agreement with spatially adjacent cell types" Methods). In this case, we made use of the single-cell resolutions of these datasets to identify both the spatially adjacent cell types and to obtain the interaction predictions. For the seqFISH dataset, we found a clear association between the predicted CCC interactions and the spatial adjacency of their corresponding cell-types for NATMI, and moderate associations for logFC Mean and Connectome, while the other methods showed inconsistent trends or lack of signal (Supplementary Fig. S15). There was no trend in the merFISH dataset, likely due to the lower gene space of that dataset (Supplementary Fig. S15).

In summary, our results showed a positive association of interactions predicted by most methods and spatially-adjacent cell types in the well-structured brain cortex, while the associations were less consistent in the breast cancer subtypes. This positive association suggests that, despite the dissociation of single-cells and their grouping into cell types, CCC predictions partly reflect the expression patterns encoded by tissue spatial context.

## Discussion

The growing interest in CCC inference has led to the recent emergence of a number of methods and prior knowledge resources. To shed light on the impact of the choice of method and resource on the inference of CCC events, we built a framework to systematically combine and compare 16 resources and 7 methods, plus their consensus. We used this framework to explore in detail the content of the different resources, to compare the predictions on six different datasets when using all combinations of methods and resources, and to assess the agreement of the methods with other data modalities. Our results suggest that both the method and resource can considerably impact CCC inference predictions, and that most methods generally capture the biological signals from other data modalities.

**Resource overlap and bias**. Despite their largely common origins, different resources covered varying proportions of the collective prior knowledge. A large share of the observed overlap among resources was a result of the frequent inclusion of certain resources[8,31,33,34], particularly Ramilowski et al. [35].

When inspecting the relative compositions of the resources, we noted biases towards certain organ- or tissue-enriched proteins and functional terms. Some resources are predominantly manually-curated[8,11,16,26,27,54], while others[6,12,28,55] are composites which also import non-curated interactions. Thus, this

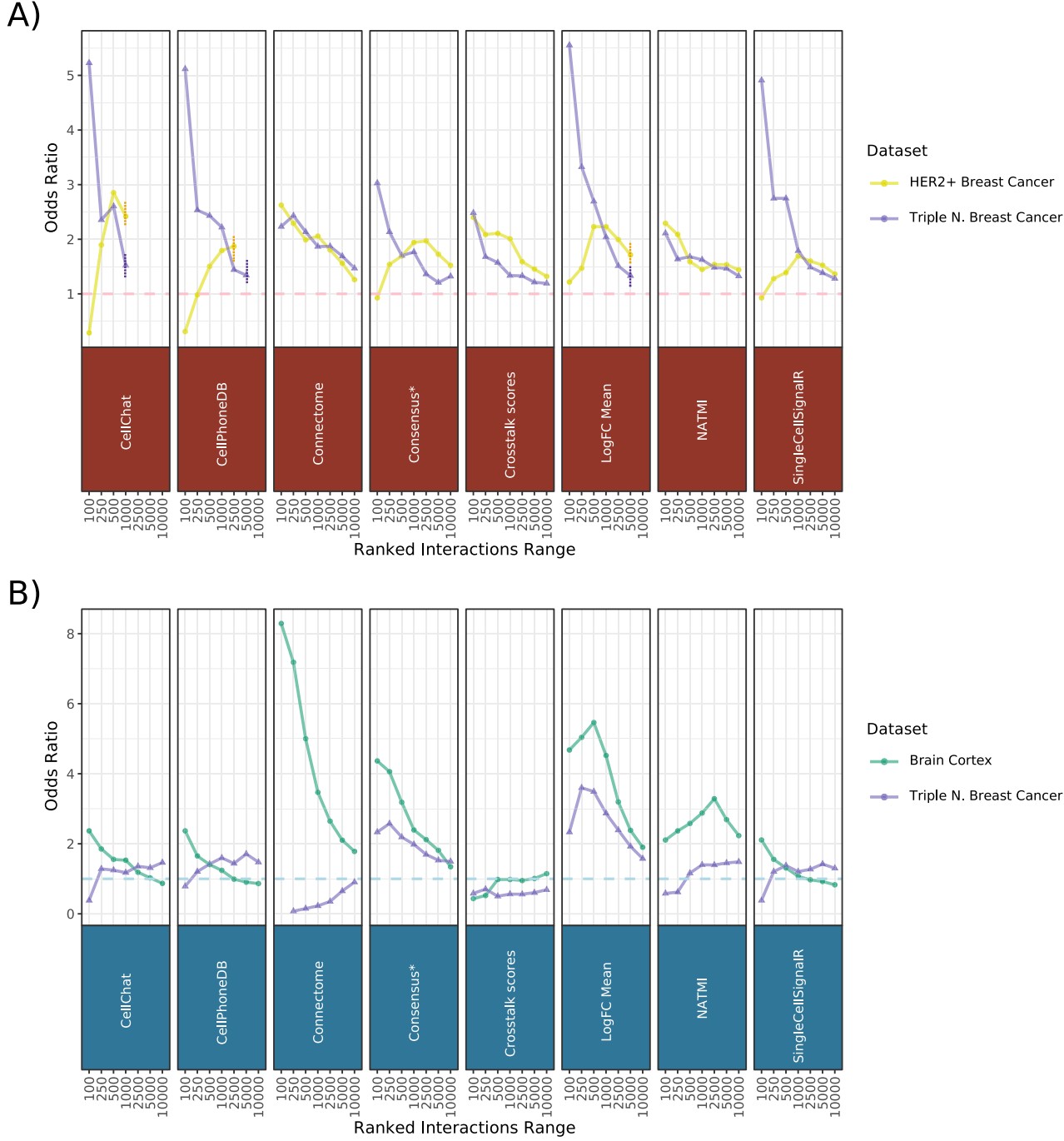

**Fig. 6 Agreement of CCC predictions with other modalities.** Odds ratios of (**A**) active cytokines and (**B**) colocalized cell types among the highest ranked interaction predictions, across a ranked range between 100 and 10,000. Odds ratios representing the association of preferentially ranked CCC predictions and (**A**) cytokine activities and (**B**) spatial adjacencies were calculated using Fisher's exact test. Asterisk (*): Consensus represents the aggregated ranks of all interactions predicted by all the methods. Dashed horizontal line is the baseline represented by an odds ratio of 1. The dashed vertical lines represent the truncated ranges of CellChat, CellPhoneDB, and LogFC Mean, arising from their relatively stricter preprocessing steps. Source data are provided as a Source Data file.

suggests a quality-coverage trade-off, as is commonly the case for biological prior knowledge. Of note, the literature-support reported by different authors for the same resources do not always agree[23,26], suggesting different interpretations of what defines a curated interaction.

These findings highlight an inherent limitation of knowledge-based inference, as any prior knowledge resource has its own biases and only represents a limited proportion of biology. Taken together, the variable overlap between the resources, their uneven

functional distributions, and the reported curation disagreements are a call for further large-scale curation efforts.

**Impact of methods and resources.** Our systematic analysis using different combinations of resources and methods revealed that both had a considerable effect on the predictions. In the case of the resources, the disagreements were largely expected as a consequence of their varying overlap. However, this was not necessarily the case for the methods, given their conceptually common

aim, similar assumptions, and previously reported agreement among some of them[10–12,14].

A major reason for the low overlap between the methods was their distinct approaches to identify the most relevant interactions. Hence, the common practice of using the number of interactions reported between two cell types as a proxy for their communication intensity is likely biased by the choice of CCC inference method. Reassuringly, our robustness analyses highlighted that the methods are fairly robust to cluster subsampling, as well as the introduction of noise to both the dataset and the resource. Collectively, these results indicate that while the methods are fairly robust to technical noise, the choice of method and resource is likely to have a major impact on the results. Therefore, downstream analyses and biological interpretation of the predicted ligand-receptor interactions should be considered with caution.

**Agreement with other data modalities**. Motivated by the observed discrepancies, we supported the methods' performance using complementary data modalities. We found concordance of the CCC predictions with receptor protein specificity and with cytokine activities estimated from downstream gene-expression signatures[48]. Of note, the cytokine activities and receptor proteins, presented in this work as an evaluation, could also be used to improve the confidence in predictions[56]. Similarly, other analyses such as pathway[14] or transcription factor activities[15,57], as well as other types of cell-communication dedicated methods, including NicheNet[19], CytoTalk[22], and SoptSC[20], could be utilised to provide further confidence in the predicted ligand-receptor interactions.

Furthermore, similarly to previous efforts, we used spatial information to support the methods' predictions[14,22]. We saw that most methods prioritise interactions between colocalized cell types, and this was clearer in the well-structured brain cortex than in breast cancer tissue. These results suggest that the performance of the methods depends on the type of tissue, and that, if available, spatial information should be used to inform[58,59] or constrain[60] the predictions.

Our agreement analyses are based on assumptions that are only approximations of reality. The limitations include the restricted coverage of the cytokine activity signatures and receptor proteins, and the technical shortcomings of current spatial transcriptomics technologies. Furthermore, such benchmarks cannot distinguish simple co-expression from actual CCC events, and do not capture complex relationships between CCC events. Since a gold standard is currently not available and the biological ground truth is largely unknown[2,25], our analyses cannot give a definitive answer of what method is best. However, we believe that these results are useful to indirectly support the methods' predictive potential.

Overall, our results suggest that despite their relatively low agreement, the CCC methods are generally able to capture relevant biological signals, and that leveraging information from additional modalities and analyses could help to refine the predictions.

**CCC inference assumptions and limitations**. The shared purpose of the methods considered in this work is to predict the most relevant interactions, commonly between a secreted ligand and its receptor, each expressed by a particular cell type. All methods work under the assumption that the expression of a pair of genes at the cell type level is informative of CCC events. Some of the methods such as CellChat[14], CellPhoneDB[8], and others[16,27,28], go a step further by considering heteromeric complexes. Ensuring that all subunits of a protein complex are expressed to consider a cell-cell interaction valid has been shown to reduce false positive predictions, and can thus impact significantly downstream interpretation and validation[8,14]. CellChat additionally accounts for interaction mediator proteins[14]. Another common assumption among the CCC methods is that cell-type-specific interactions are more informative than those shared by multiple cell types[8,10,11,14]. Yet by focusing on the cluster-specific interactions, the predictions may not capture biologically relevant processes that are common between multiple cell types[12].

Gene expression provided by scRNA-Seq is typically limited to the cells within the dataset, and hence does not capture long-distance endocrine signalling events. In addition, CCC inference from scRNA-Seq data assumes that gene expression of a transmitter and a receiver is a good proxy for their joint activity, without considering any of the processes preceding transmitter-receiver interactions, including protein translation and processing, secretion, and diffusion[2]. Furthermore, gene expression is a proxy of protein levels alone, yet recent efforts attempt to capture signalling events mediated by other molecules such as neurotransmitters[15,16]. Finally, current methods are limited to single species although some information about interspecies communication can be inferred[61,62].

## Conclusions

Considerable efforts have been made to develop CCC inference tools and resources, and we expect that further advancements will be key for the systems-level analysis of single-cell data. The popularity of CCC inference is anticipated to increase as spatial transcriptomics[1] and single-cell proteomics[63] continue their rapid development. We regard the results presented here as steps towards an understanding of the strengths and weaknesses of CCC methods, and LIANA as a framework for their further analysis, benchmark, use and development.

## Methods

**Processing of CCC resources**. The connections between resources shown in the dependency plot were manually gathered from the publications and the web pages of each CCC resource.

OmniPath is a comprehensive knowledge database with more than 100 intracellular and intercellular resources[3]. The OmniPath intercellular component is a composite resource which contains interactions from all of the CCC dedicated resources compared here, along with some additional resources[3]. All the CCC resources used in the analyses presented in this work were queried from OmniPath[3], with the exception of CellCall which was processed to OmniPath format separately. The contents of the resources are identical to their original formats, apart from minor processing differences (Supplementary Table 2), such as removal of duplicates, updating to the latest gene symbols, or removal of genes lacking reviewed Uniprot IDs. All complex-containing resources were dissociated into individual subunits for the resource-focused analyses presented in this work.

OmniPath's version used in this work was filtered according to the following criteria: (i) we only retained interactions with literature references, (ii) we kept interactions only where the receiver protein was plasma membrane transmembrane or peripheral according to the 51st consensus percentile of the localisation annotations, and (iii) we only considered interactions between single proteins (interactions between complexes are also available in OmniPath). Tutorials on how to customise OmniPath as well as how to make use of the intracellular functional information available at OmniPath are available at https://saezlab.github.io/liana/. OmniPath's intra- and intercellular components were both obtained and are both available via the OmnipathR package (https://github.com/saezlab/OmnipathR).

**Descriptive analysis of resources**. We defined unique and shared interactions, receivers and transmitters between the CCC resources if they could be found in only one or at least two of the resources, respectively.

To identify uneven distributions of transmitters, receivers, and interactions toward biological terms or protein localisations, we used Fisher's exact test to compare each individual resource to the collection of all the resources. The test $p$-values were FDR corrected. We performed the analysis using the aforementioned functional annotation databases in 3 distinct categories. For the overrepresentation of interactions, we considered annotations when both the transmitter and receiver were matched to the same category, while annotations matched to transmitters and receivers enrichments were examined independently. We allowed the same protein or interaction to be matched to multiple pathways or functional categories from the same database. Interactions, receivers, and transmitters were independently

matched to the 10 pathways from SignaLink[38], and the 15 largest categories from CancerSEA[41], and NetPath[40]. The same procedure was also applied to organ- and tissue-enriched proteins from the Human Protein Atlas[42], accessible at https://proteinatlas.org, and disease-associated genes from DisGeNet[43]. Pathology-associated, uncertain, and unsupported proteins with a low/non-representative level of expression were excluded from Human Protein Atlas database, while DisGeNet gene-disease associations were filtered to include only literature-supported associations (GDA Score > = 0.3). Each of the aforementioned general functional annotation databases was obtained via OmniPath and their protein complexes, if present, were also dissociated.

We also obtained protein localisations from OmniPath which collects this information from 20 databases[3]. Then we kept consensus protein localisations above the 51st percentile. We classified CCC interactions using the localisation combinations of proteins involved in the interactions, which included secreted, plasma membrane peripheral and transmembrane proteins.

**Input specifics**. For the method-resource comparisons and evaluations, we used Seurat[46,64] objects which were converted to the appropriate data format when calling each method. Whenever available, we used the recommended conversion method or wrapper for each method. Log-transformed counts were used when this was not done internally by the method.

The complex-containing interactions, if present in a given resource, were dissociated for the methods which do not take complexes into account, namely the original implementations of NATMI, SingleCellSignalR, and Connectome.

**Method specifics**

*CellChat*. CellChat was run using its default settings with 1000 permutations and the gene expression diffusion-based smoothing process was omitted.

*CellPhoneDBv2*. CellPhoneDB's algorithm[8] was re-implemented in LIANA and used throughout this manuscript with 1000 permutations. Identical to the original implementation, cluster labels were reshuffled and an one-sided empirical $p$-value was calculated for the interactions with a mean expression higher than random. Only interactions whose transmitter and receiver genes were expressed in at least 10% of the cells were considered, and the subunit with the minimum expression was used for complexes.

*Connectome*. Connectome was run with its default settings and filtered for differentially expressed genes ($p$-value < = 0.05), as identified via a Wilcoxon test.

*logFC Mean*. The LogFC Mean score implemented in LIANA, was inspired by iTALK[6], and it represents the average of one-versus-the-rest log2FC expression changes for the transmitter and receiver cell types. The logFC Mean score uses LIANA's default filtering settings, namely both the transmitter and receiver genes of any interaction evaluated must be expressed in at least 10% of the cells, and it considers the subunit with the minimum expression for complex-containing interactions.

*SingleCellSignalR*. SingleCellSignalR was run with the processed gene counts, considering differentially expressed genes with a log2 fold change threshold of 1.5 or above, and we filtered LRscores > = 0.5 for the evaluations. The "int.type" parameter was set to "autocrine". We noted that this option returned both paracrine and autocrine signalling interactions. The source code of SingleCellSignalR was modified to work with external resources (available at https://github.com/saezlab/SingleCellSignalR_v1).

*NATMI*. NATMI's implementation is command-line based, thus a system command is invoked via R that calls the NATMI python module and passes the appropriate command line arguments. NATMI was run with its default settings using the processed gene expression matrix, converted from Seurat. The source code of NATMI was modified to be path-agnostic and to work with integers as cluster names (available at https://github.com/saezlab/NATMI).

*Crosstalk scores*. Crosstalk scores, inspired by CytoTalk[22], were implemented in LIANA. CytoTalk's crosstalk scores are composed of two metrics: the preferential expression measures (PEMs) and the non-self talk scores (NSTs). The first one reflects the specific expression for quantified genes across all the cell types. The latter is defined on the basis of information theoretic measures and quantifies the mutual information (Shannon entropy) for a pair of genes (ligand and receptor) within the same cell type, and is thus designed to penalise autocrine signalling. Once NST and PEM are calculated, the crosstalk score is calculated for each ligand-receptor pair and for each cell type pair as the product of the minmax normalised PEM and NST values. To enable the comparison to the rest of the methods, and in contrast to the crosstalk scores implemented in CytoTalk, we calculated the crosstalk scores by cell type pairs and used the inverse of the non-self-talk scores for autocrine signalling interactions. Moreover, our implementation considers complexes, and interactions with transmitters or receivers with preferential expression measures of 0 are also assigned 0.

*Robust-rank aggregate*. A consensus rank is generated across all methods using Robust Rank Aggregation[65]. These aggregated ranks can in turn be interpreted as a probabilistic distribution for interactions that are preferentially highly-ranked. The aggregate ranks are built across the universe of all interaction predictions, after independent filtering by each method. By default, missing interactions are imputed as the max ranks.

**Overlap analysis**. To compare the overlap between the interactions predicted by each method-resource combination, we kept the 1,000 highest ranked interactions by default, including ties. We also considered the highest ranked 1% of interactions for each method, including ties. We then generated a presence-absence matrix of predicted interactions with method-resource combinations. These matrices were subsequently used to calculate the reported Jaccard indices.

Unless explicitly mentioned, and if available, we used the scoring functions for each method recommended for single-condition interaction predictions (Supplementary Table 3).

Frequencies of interactions per cell type were calculated using the highest ranked hits for each method-resource combination. These frequencies represent the proportion of top predicted interactions (or edges) that stem from or lead to a source or target cell type, respectively. In other words, interaction frequencies represent the relative number of interactions per cell type within the highest ranked 1000 interactions.

The relative interaction strength by cell type was calculated using the regularised scores from each method, i.e. all scoring functions were scaled between 0 and 1. Then the mean regularised score per cell type, categorised as source or target, was divided by the average score of all interactions predicted.

**Agreement with other modalities and robustness**. All of the comparisons with other modalities were performed using the OmniPath CCC resource. For murine datasets, we converted the OmniPath to murine symbols using the biomaRt package[66].

For the binary categorisations used in the agreement with cytokine activity analysis and spatial adjacencies, we performed Fisher's exact test, sequentially in rank intervals ranging from 100 to 10,000, to obtain the Odds ratios of the positive and negative classes against a background universe. In the case of the spatial adjacency analysis, the background universe contained all predicted interactions, while for the cytokine activities, we only considered those matched to cytokines from CytoSig[48].

*Agreement with cytokine signatures*. CytoSig provides a collection of consensus, data-driven, cytokine-activity signatures compiled using a compendium of transcriptomic profiles[48]. We used CytoSig's 43 high-quality signatures to infer which cytokines induce signalling activities in each cell type. We then used this information to assess if a cytokine-receptor interaction reported by the different CCC methods was supported by the corresponding cytokine downstream signalling activities.

We computed the cytokine activity scores for all cell types with the multivariate linear regression model ('mlm') method of decoupleR at the pseudobulk level. We chose the mlm method as an approach that models the effect of multiple cytokines and that performed best in a recent footprint-focused analysis benchmark[67].

To build the pseudobulk profiles, we log2-transformed the summed counts within each cell type, and kept only genes which were expressed in at least 10% of the cells and with a summed raw count above 5.

In this evaluation, we used both the autocrine and paracrine CCC predictions, calculated using expression counts at the cell-type level for all cell types, from the HER2 + and triple negative breast cancer subtype datasets[44]. We considered any cytokine signature with a positive score and FDR-corrected $p$-value = < 0.05 in the target cell types as an active cytokine. We considered all CCC predictions with a ligand corresponding to a CytoSig signature, including the same ligand to multiple receptors, matched to any of the aliases of the cytokines. Odds ratios were then calculated as the ratio between any CCC prediction with corresponding active cytokine in a given receiver cell type, and those assigned to the negative class—i.e. the remainder of the cytokine signatures.

*Agreement with spatially adjacent cell types*. We used the SPOTlight[68] deconvolution method with default parameters to spatially map the cell types present in our scRNAseq datasets into their corresponding 10× Visium slides. SPOTlight provides cell type proportions per spot that were subsequently used to identify colocalized cell types by computing Pearson's correlation. The Pearson coefficients were scaled to create a distribution of correlations, and only considered the most strongly correlated cell type densities ($z$-score > = 1.645) as colocalized, while the remainder of the cell pairs were considered as non-colocalised.

The mer- and seqFISH datasets were already annotated and provide single-cell spatial resolution, hence the same dataset was used to obtain CCC predictions and spatial information. To identify the enriched neighbouring cells for each cell type mer- and seqFISH datasets, we used Squidpy's[69] Neighbourhood Enrichment analysis with its default parameters. In accordance with the approach followed with the 10× VISIUM slides, we considered significantly colocalized cell type pairs with a normalised neighbourhood enrichment score > = 1.645 as spatially adjacent.

*Agreement with receptor protein abundance.* To identify specifically expressed receptors across clusters, we z-transformed receptor protein abundance across cell types. Receptors with an abundance *z*-score > = 1.645 were considered specifically abundant at the protein level. These receptors were then treated as the positive class, while all others were assigned to the negative class. AUROC and AUPRC metrics were calculated using yardstick[70]. For the AUPRC calculations, we downsampled the negative class 100 times to match the (lower) number of receptors assigned to the positive class. The downsampling procedure binds the expected random AUPRC to 0.5.

We allowed surface protein receptors to match multiple genes (e.g. T-cell receptors subunits), and vice versa. Gene aliases of proteins were obtained using the human and mouse gene databases from the org.Hs.eg.db[71] and org.Mm.eg.db[72] BioConductor packages. Proteins with non-standard names, or absent aliases in the aforementioned databases, were manually annotated using UniProt[73] as a reference.

*Robustness analyses.* To evaluate sensitivity of the methods to noise, we performed four distinct robustness analyses. We simulated noise in the data by subsampling the number of cells per cluster and by reshuffling the cell type labels.

Additionally, to simulate the impact of false interactions in the resource, we randomly generated interactions from the 2000 most variable genes in the dataset and randomly replaced proportions of the resource with these putative false interactions. In one scenario, we selectively replaced interactions in the resource and preserved the highest ranked interactions, while in the other scenario we non-selectively swapped any of the interactions.

All four analyses were done in an iterative manner over a range of manipulations (0–40%). We treated the highest ranked 250 interactions from the non-modified resource/data as ground truth and repeated the randomisation process 5 times.

**Data processing**. All 10× Genomics, including all CITE-Seq and the 3k PBMC, datasets were processed using the standard Seurat pipeline. Namely, filtered gene expression count matrices were log-normalised, and if cell type annotations were not provided, the cells were clustered, following scaling, identification of variable features, and PCA dimensionality reduction, using Seurat's[64] (v4.0.3) default settings. For 10× Genomics CITE-Seq datasets we used a clustering resolution of 0.4 and the protein abundances were centred-log-ratio transformed. In the Murine spleen-lymph CITE-Seq datasets[74], duplicated and low quality cells, as annotated by the original authors, were filtered, in agreement with the other CITE-seq datasets, gene counts were log-normalised, while protein abundances were centred-log-ratio transformed.

For the colorectal cancer dataset, we kept the original subtype labels, reformatted the names to work with each CCC method, and sparsified the counts into a Seurat[64] object. The pre-processed and labelled Pancreatic islet[46] and cord blood mononuclear cell[45] datasets were log-normalised, and subsequently used for CCC inference. In the latter dataset, any murine and doublet/multiplet cells, as annotated by the authors, were excluded.

We used ComplexHeatmap[75] to generate the heatmaps and ggplot2[76] for any of the other plots presented in this work.

**Reporting summary**. Further information on research design is available in the Nature Research Reporting Summary linked to this article.

## Data availability

The processed and annotated Human Breast Cancer single-cell atlas[44] is available via the GEO accession number: GSE176078. The filtered breast cancer 10× Visium slides from the same publication are available at https://zenodo.org/record/4739739. Processed seqFISH[77] [https://content.cruk.cam.ac.uk/jmlab/SpatialMouseAtlas2020/] and merFISH[53] (GEO accession number: GSE113576) datasets were obtained via the spatial single-cell analysis framework—Squidpy (v1.1.0)[69] [https://squidpy.readthedocs.io/en/latest/api.html#module-squidpy.datasets].

Pancreatic islet[46] (GEO accession numbers: GSE84133, GSE81076, GSE85241, GSE86469; ArrayExpress: E-MTAB-5061) and cord blood mononuclear cells[45] (GEO accession number: GSE100866) scRNA-Seq datasets were obtained via SeuratData (https://github.com/satijalab/seurat-data).

Publicly available 5K PBMC, 5K PBMC NextGem, 10K PBCM, and 10K MALT CITE-Seq datasets were obtained from 10× Genomics (accessible under the list of datasets at https://tinyurl.com/10xCITEseq).

Processed and annotated murine spleen-lymph CITE-Seq datasets[74] are available via the GEO accession number: GSE150599.

The processed single cell RNA-Seq data[47] for 23 Korean colorectal cancer patients are available via the GEO accession number: GSE132465.

Spatial transcriptomics datasets (10× Visium slides) on sagittal adult mouse brain anterior and posterior slices were obtained from SeuratData, available at https://github.com/satijalab/seurat-data, under the dataset name of 'stxBrain', or publically via the 10× Genomics website under Spatial Gene Expression v1 Chemistry datasets [https://tinyurl.com/10xVisiumDemonstration]. The single-cell data (Allen Brain Atlas[51]) used for the cell type mapping (deconvolution), was obtained as a Seurat object, accessible at https://www.dropbox.com/s/cuowvm4vrf65pvq/allen_cortex.rds?dl=1, and is alternatively available via accession number: GSE71585.

The 10× Genomics' 3k PBMC dataset used in the robustness analysis is available at https://cf.10xgenomics.com/samples/cell/pbmc3k/pbmc3k_filtered_gene_bc_matrices.tar.gz. Source Data for all Supplementary Figures, along with preprocessed outputs, are available at: https://zenodo.org/record/6531218. Source data are provided with this paper.

## Code availability

The LIANA framework is available at https://github.com/saezlab/liana, and the version used to generate the results presented here is available via Zenodo[78]. The scripts used to generate the results presented here can be accessed at https://github.com/saezlab/ligrec_decouple.

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

## Acknowledgements

This work was supported in part by the European Union's Horizon 2020 research and innovation program (860329 Marie-Curie ITN "STRATEGY-CKD") and the German Federal Ministry of Education and Research (Bundesministerium für Bildung und Forschung BMBF) Computational Life Sciences LaMarck grant no. 031L0181B), awarded to J.S.R. This work was in part funded by the clinical research unit CRU344 supported by the German Research Foundation (DFG) and the E:MED Consortia Fibromap funded by the German Ministry of Education and Science (BMBF) awarded to I.C. We express our gratitude to Erick Armingol, Pau Badia i Mompel, Hratch Baghdassarian, Luz Garcia-Alonso and Suoqin Jin for their helpful feedback and discussions and to Ece Kartal for the design of LIANA's outline graphic. For the publication fee we acknowledge financial support by Deutsche Forschungsgemeinschaft within the funding programme "Open Access Publikationskosten" as well as by Heidelberg University.

## Author contributions

J.S.R. conceived the project. D.D. set up the framework used in this manuscript, with the help of D.T., M.G.R., and J.S.N. D.D. performed the comparisons and evaluations

presented in this work with the support of A.D., A.V., R.O.R.F., and J.S.R. D.T. set up the resource formatting infrastructure with the help of D.D. D.T., D.D., and C.B. created the resource analysis pipeline. P.L.B. performed the robustness analysis under the guidance of D.D. J.S.R. supervised the project with the help of A.V. and A.D. H.K., R.O.R.F., and B.S. performed preliminary and supplementary analyses that helped shape the work presented here. I.C. supervised J.S.N. A.D. and A.V. contributed equally to the manuscript. All authors contributed and revised the final version of the manuscript.

## Funding

## Competing interests
J.S.R. has received funding from GSK and Sanofi and fees from Travere Therapeutics. A.V. is currently employed by F. Hoffmann-La Roche Ltd. The remaining authors declare no competing interests.
