## [Peer review file. · Nature Communications]

REVIEWER COMMENTS

Reviewer #1 (Remarks to the Author): Expert in cell-cell communication methods, single-cell RNA-seq, and bioinformatics

An increasing number of tools have been recently developed for inferring cell-cell communication by combining single cell RNA-seq (scRNA-seq) data with prior curated ligand-receptor interaction resources. Accurate inference is thus crucial to biological interpretation and understanding cellular behavior. In this manuscript, Dimitrov et al. developed a general framework to systematically compare 15 resources and 6 methods. This work allows more systematic exploration of the impact of the choice of resources and methods on the resulting predictions. Using a colorectal cancer scRNA-seq dataset, a systematic analysis revealed major differences in the built-in resources and the inferred interactions among different methods. This is a very timely comparison study, highlighting significance of different resources and methods on the biological interpretations, and the need of a further benchmark study. The general framework, implemented as a R package, is also helpful for future development of new methods and further comparison study.

On the details of comparisons, I've a few comments.

Comments:

1. The authors categorized interactions into secreted, direct-contact and Other in the section 2.2.1. It is unclear what kind of interactions are considered as 'Other'. Can the author provide several representative examples? It is helpful to have a table showing a couple of examples in each category.
2. The authors stated 'a reshuffled control resource' in the section 2.3. However, the aims and conclusions of this related analysis are unclear. How did the authors reshuffle the resource? Did the authors randomly shuffle the pairs of ligands and receptors to create a fake resource? If the resource contains wrong ligand-receptor pairs, can these methods filter out these false positive interactions?
3. The authors defined the cell type activities as the proportion of interactions per cell type. Such definition ignores the interaction strength that is computed based on the expression of ligands and receptors. Does this mean that the number of interactions is a more biologically meaningful measurement for characterizing cell type activities compared to the interaction strength individually or in total?

4. In running of Connectome, the scaled weights were used. Why not use the normalized weight, which approach is a more general way when quantifying the interaction strength?

5. OmniPath integrates all the resources together and provides a comprehensive resource to the community. Thus, the quality of the interactions in OmniPath is very important for correct biological interpretations. Does OmniPath include a confidence score or literature to allow users select high confidence interactions? In addition, the authors categorized the interactions into functional terms such as innate immune pathways. Is such information available in OmniPath? It is useful to have it so that users may select a particular set of interactions, accordingly.

6. In Supplementary Table 1, is the number showing in the table the shared interactions or unique ones?

7. What is the meaning of each row in Figure 4?

8. On page 11, 'CellChat' should be 'CellChatDB'

Reviewer #2 (Remarks to the Author): Expert in cell-cell communication methods and single-cell RNA-seq

This paper first performed a comparative assessment of known ligand-receptor (LR) resources and then compared all combinations of 15 resources of LR interactions and 6 methods of inferring cell-cell communications (CCC) on a colorectal cancer scRNA-seq dataset. As far as I could understand, the author compared the dependencies and overlap between 15 resources and localization and functional term abundance in CCC resources. On the other hand, the author compared the 500 highest ranked CCC interactions between different combinations of methods and resources. Unfortunately, the comparisons are shallow and biased lacking deep and systematic evaluation as well as informative conclusions on the CCC resources and methods, which makes it hard to see the novelty of this work. It is easy to understand that different combinations are supposed to provide different results as the algorithms underlying methods and the LR pairs underlying CCC resources are different, wherein this work only showed the different results of inferred LR pairs without a comprehensive discussion of the strengths and weakness of each method/resources and a reasonable explanation of the observed disagreement.

Specific comments:

1. As mentioned above, the comparisons in Results section are too shallow. More methods (e.g., CytoTalk, NicheNet, SoptSC) are required to be compared. Although these methods can infer the downstream activated pathways different from the CellChat, CellPhoneDB, SingleCellSignalR, etc., they are the representative CCC methods in this area, which have demonstrated an appealing performance in their benchmarking analysis. In addition, more datasets are required to be benchmarked to generate an unbiased comparison.
2. Indeed, there is no gold standards in this area to evaluate the CCC resources and methods. However, there are some efficient strategies to evaluate the methods' performance which are widely-used as the authors stated in the manuscript, including a presumed correlation between CCC activity and spatial adjacency, recovering the effect of receptor gene knockouts, robustness to subsampling, agreement with proteomics, simulated scRNA-Seq data, and the agreement among methods. The author also should apply these strategies and the corresponding datasets (e.g., spatial datasets, scRNA-seq datasets with the knockout of ligands/receptors, single-cell multimodal datasets, simulated scRNA-seq datasets) to generate a systematic evaluation.
3. In general, it's a descriptive research. A comprehensive discussion of the strengths and weakness of each method/resources and a reasonable explanation of the observed disagreement should be provided and more informative and unbiased conclusions should be drawn based on the enlarged benchmarking datasets and comprehensive comparison.

Reviewer #3 (Remarks to the Author): Expert in cell-cell communication methods, single-cell RNA-seq, and cancer genomics

Summary

The manuscript describes and compares global statistics of the existing resources and tools to infer cell-cell communication (CCC) events from scRNAseq experiments. It represents a comprehensive overview of the existing CCC resources and strategies. It attempts to compare the many published CCC frameworks, a need for the scRNAseq community.

Major comments

While the question "how the existing CCC approaches compare?" is extremely relevant, the comparison performed is far from providing guidance in choosing a CCC approach. The manuscript needs additional

work, stating and assessing more useful questions and reducing redundant and nonessential statistics provided. Generally, while summary statistics of the resources and tools are informative, the need is more on identifying the approach able to prioritize the CCC key for a tissue, disease or developing process, that can be reproduced and validated in the laboratory. We agree that such an objective is challenging due to few available datasets and the complexity of the questions. Still, we encourage the authors to emphasize that the value of a resource is not a function of the number of interactions reported, but how well these represent experimentally validated interactions with clear role in cell-cell communication. Similarly, the value of the tools should not be measured by the similarity of their results, but by their ability to help the researchers to identify the CCC driving a disease or a differentiation process.

Please find below a series of suggestions and points that need clarification.

1) Resource Uniqueness and Overlap. While the total number of interactions and their overlap can be informative, the more interactions and their uniqueness is not a measure of quality. Especially when comparing a) resources that uniquely contain their own “expert curated” interactions (such Ramilowski2015/NATMI, CellphoneDB or HPMR) to b) other resources that import interactions from (a) or include interactions that have not been curated by experts. We suggest the authors distinguish between expert curated and non-expert curated interactions. A curated interaction is that with a supported role in cell-cell communication (i.e. induces a response in the interacting cells or has a structural role - e.g. adhesion molecules). An interaction between a secreted protein and a membrane bounded protein identified by co-immunoprecipitation or yeast-two hybrid is not a curated interaction as their role in cell-cell communication is not experimentally supported. Other pertinent questions are:

- Out of the total number of candidate CCC interactions reported by all these resources: how many interactions are supported by experts or have a proven role in cell-cell communication (or alternatively how many are potentially novel candidates)? This will help the reader to understand the size of the bona fide or candidate CCC.

- What resources are contributing to each type of interaction? This can help the reader to understand the risks (or the likelihood of finding novel CCC) in choosing a resource.

- What % of curated interactions are borrowed from other resources? Which resource incorporates the largest number of non-borrowed expert-curated CCC? This information can be included as edge width in figure 2a, and will acknowledge the contribution of each resource to the CCC census.

- The amount of unique CCC reported by Omnipath and CellChatDB is worrying. What is the evidence for these interactions? What are the proteins that more frequently appear in the Omnipath and CellChatDB unique categories? Can the authors provide these?

2) The classification of proteins as receivers or transmitters is confusing. Except for CCC involving secreted ligands, the remaining interactions likely have a bi-directional effect on the interacting cells.

Are adhesion molecules, such as integrins, reported as receivers or transmitters? Are these ignored or counted twice?

3) For resources considering complexes, how are interactions quantified and compared to the resources ignoring the complex structure? Are the complexes decomposed in pairs (for example is simple(A) <-> complex(B1+B2) decomposed as A-B1 and A-B2 rendering two pairs of interactions) or are they kept as a unit (A-B1/B2)? This is important because for two resources with the exact same information, the one ignoring the complex structure will systematically have more interactions.

4) We don't find that the section "2.2.1 Subcellular Localisation" provides critical information to choose a resource. This section can be shortened. It is expected that the majority of resources will be biased towards secreted proteins and membrane bound receptors as these are the most obvious CCC candidates for both manual curation and in silico predictions. Pharmacology, for obvious reasons, will be biased towards secreted proteins.

5) Also, not sure what the analysis with "Functional cancer cell states from CancerSEA" adds here. It is hard to predict what impact "DNA damage states were over-represented in LRdb " will have, as the link between cell-cell interactions and DNA damage is ambiguous. A more useful information would be to know the bias towards particular tissues/diseases. For example, if I am studying the brain, which is the resource covering larger % of brain-specific proteins (and the % of curated CCC)? This is also relevant for immune studies. Immune cells are better characterized because of tissue accessibility. Resources that uniquely incorporate manually curated CCC may likely be biased towards immune CCC. One could use GTEx or human atlases to identify tissue or lineage-specific genes.

6) Regarding the "Agreement in CCC predictions", the authors "looked at the overlap between the 500 highest ranked interactions as predicted by each method". Can Table 1 be extended to include what measurement from each tool they use as a ranking score? Are these scores a measure of specificity (cluster-enriched), mRNA abundance (avg. expression), etc?

7) The fact that the results depend on the method and not on the resource is unexpected. Can the authors elaborate more on the reasons why this can be the case? For example, are the methods sensitive to the size of the clusters? Do all the methods normalize the input data so that measurements are comparable within the dataset?

8) The "interaction overlap" between methods, although useful to highlight the differences between approaches, does not inform on their ability to retrieve relevant biology. In the absence of a gold standard, it may be useful to quantify the enrichment in cluster-specific genes in the 500 highest ranked interactions predicted by each method. This can be done with classical differential gene expression

(DGE) analysis or alternative methods looking for specificity, such as TF-IDF. Cluster-specific genes will be a constant in each annotated dataset and could be a good reference to compare methods under the same resource. A CCC tool that misses most of these genes (with a CCC role) may not be that appropriate. It should be noted that CellChat and iTALK already rely on DEGs.

9) The biological value of annotating the complexes should be highlighted in the discussion as ignoring such nature will generate false positives, no matter the tool used. For example, if a receptor is a heterodimer and one unit is not expressed, the CCC is just not possible. Instead, the authors mention “resources, such as Baccin, CellChatDB, CellPhoneDB, and ICELLNET, take protein complexes into account. This largely complicates the conversion of the resources”. Again, the focus should be on the quality of the predictions and not on the comparability of the resources, especially when simplification induces false signals. Resources can be compared if these are decomposed into the same units.

10) The authors use the word “activity” several times in the text. Activity implies signal transmission and is a concept that only methods accounting for downstream signaling actually explore (such NicheNet). What do the authors mean by “CD4+ T and CD8+ T cells ... were assigned a CCC communication activity.” How are the authors quantifying CCC activity? In the same sense, what does the sentence “actively communicating cell clusters” mean? A single CCC interaction may be enough to have an “active” communication between two cells (for example a strong ligand inducing cell death). If the authors use activity as a function of the “number of CCC reported between two cell types”, we strongly suggest they rephrase the text to avoid confusion.

Minor comments

- The text should be shortened for effectiveness. Most of the statistics described do not add but distract the reader from the main message.

- CellphoneDB is a tool as well (Figure 1). Actually, CellphoneDB is among the leading tools in the field (with ~300 citations), while Squidpy independent implementation of this tool has not been peer reviewed yet. CellphoneDB's original tool should be included in the comparison.

- Why are the authors not including the tool NicheNet? We agree that NicheNet is not directly comparable as it accounts for downstream signalling, but at least should be mentioned and listed as a CCC method.

- Sentence “By focusing on the cluster-specific interactions in the dataset, these methods report the most specifically-interacting cell types”. The methods do not report interacting cell types but the interacting proteins between two cell types. What is tested for specificity is the protein/cell type combo. An alternative could be “By focusing on the cluster-specific interactions in the dataset, these methods report the interacting proteins used specifically/exclusively by a pair of cell types”.

- In the sentence "Such an assumption inherently suggests that gene expression is informative of the activity of transmitters and receivers. However, gene expression provided by scRNA-Seq is typically limited to protein coding genes and the cells within the dataset". "However" should be replaced by "also, additionally" as sentences are independent/additive.

Legend:

Response

Reference to Main text

REVIEWER COMMENTS

Reviewer #1

(Remarks to the Author): Expert in cell-cell communication methods, single-cell RNA-seq, and bioinformatics

An increasing number of tools have been recently developed for inferring cell-cell communication by combining single cell RNA-seq (scRNA-seq) data with prior curated ligand-receptor interaction resources. Accurate inference is thus crucial to biological interpretation and understanding cellular behavior. In this manuscript, Dimitrov et al. developed a general framework to systematically compare 15 resources and 6 methods. This work allows more systematic exploration of the impact of the choice of resources and methods on the resulting predictions. Using a colorectal cancer scRNA-seq dataset, a systematic analysis revealed major differences in the built-in resources and the inferred interactions among different methods. This is a very timely comparison study, highlighting significance of different resources and methods on the biological interpretations, and the need of a further benchmark study. The general framework, implemented as a R package, is also helpful for future development of new methods and further comparison study.

We thank the reviewer for their positive words and we are glad that they found it relevant and helpful.

On the details of comparisons, I've a few comments.

Comments:

1. The authors categorized interactions into secreted, direct-contact and Other in the section 2.2.1. It is unclear what kind of interactions are considered as 'Other'.

RIA1. We agree that this should be better explained. We used information about the subcellular localization of transmitter and receiver proteins to categorize interactions into secreted and direct-contact. Any interactions that did not fit in the secreted or direct-contact categories were labeled as 'Other'. These include interactions for which the localization of the partners (proteins) is

undetermined, or partners are matched to multiple categories, and hence does not match any reasonable scenario (e.g. Secreted-Secreted, Transmembrane-Secreted, etc). For example, P19438 (TNFRSF1A), shown in the 7th row in Supp Table 4, is a membrane-bound receptor which is also annotated as ‘secreted’ in UniProt, due to its reported soluble form as a result of a polymorphism in multiple sclerosis (1). For further information, we kindly refer the reviewer to the expanded Supp. Note 1, and to the examples below (Supplementary table 4).

Can the author provide several representative examples? It is helpful to have a table showing a couple of examples in each category.

We agree with the reviewer that showing a few examples of each category would be helpful, thus we included the following supplementary table:

Supplementary table 4. Signalling category examples.

Signalling type	Transmitter	Transmitter symbol	Receiver	Receiver symbol	Resource
Direct-contact	Q9NQS3	NECTIN3	Q495A1	TIGIT	Cellinker
Direct-contact	P52803	EFNA5	P54764	EPHA4	Ramilowski
Direct-contact	Q9NT99	LRRC4B	Q6UXZ4	UNC5D	Cellinker
Direct-contact	P10321	HLA-C	P43632	KIR2DS4	Ramilowski
Direct-contact	Q14952	KIR2DS3	Q7Z3B1	NEGR1	Cellinker
Other	P21741	MDK	P18827	SDC1	Cellinker
Other	P01374	LTA	P19438	TNFRSF1A	OmniPath
Other	P21741	MDK	P18827	SDC1	CellChatDB
Other	P49763	PGF	P17948	FLT1	EMBRACE
Other	P06858	LPL	P13611	VCAN	Cellinker
Secreted	Q8IZJ0	IFNL2	Q08334	IL10RB	Ramilowski
Secreted	P00747	PLG	P05556	ITGB1	iTALK
Secreted	P56706	WNT7B	O75197	LRP5	Ramilowski
Secreted	P20062	TCN2	P21554	CNR1	iTALK
Secreted	P01303	NPY	Q9Y5X5	NPFFR2	ConnDB2020

2. The authors stated ‘a reshuffled control resource’ in the section 2.3. However, the aims and conclusions of this related analysis are unclear. How did the authors reshuffle the resource? Did the authors randomly shuffle the pairs of ligands and receptors to create a fake resource? If the resource contains wrong

ligand-receptor pairs, can these methods filter out these false positive interactions?

RIA2. The reshuffled resource in the first version was created by reshuffling the edges of ConnectomeDB2020 using BiRewire (2), which swaps ligands and receptors preserving the overall network properties, indeed creating a ‘fake’ resource.

We agree that the aims and conclusions of the reshuffled resources were not very informative and we have replaced this analysis with one that specifically assesses the robustness of the different methods to introducing putative wrong interactions to the resource. This was done by generating interactions between random genes, reflecting what the reviewer suggests by adding false positive interactions. Our analysis represented the addition of ‘novel’ and replacement of non-relevant interactions to a resource (Supplementary Figure S11C).

Our robustness analysis suggested that the methods are fairly robust to changes in the resources, as even in the extreme cases where we introduce large proportions of false interactions (40%), the methods recovered most of the true positive highly ranked interactions.

Supplementary Figure S11. Robustness to [...] C) Selective replacement of interactions with non-canonical ones [...]. The non-modified predictions (0% modifications) were considered as the ground truth to calculate the Jaccard indices.

3. The authors defined the cell type activities as the proportion of interactions per cell type. Such definition ignores the interaction strength that is computed based on the expression of ligands and receptors. Does this mean that the number of interactions is a more biologically meaningful measurement for characterizing cell type activities compared to the interaction strength individually or in total?

RIA3. This is a valid point from the reviewer, and indeed the number of interactions is not necessarily more meaningful than the strength. Thus, we computed the relative interaction strength per cell type based on the interaction scores returned by the different methods (Supp. Figs 21 and 22), represented by the average score assigned to each subtype relative to the mean score across all cell types. This way we take into account the strength and also consider all predicted interactions, not just the highest ranked interactions.

4. In running of Connectome, the scaled weights were used. Why not use the normalized weight, which approach is a more general way when quantifying the interaction strength?

RIA4. This is a valid question as both metrics provide useful information. The authors of Connectome's paper state:

“This edgeweight [**weight_scale**] is meant to be used when exploring a single cell system [...]” (3).

“Weight_{norm} [**normalized weight**] is useful for differential connectomics, in which exact edges are compared across tissue conditions.” (3)

Based on this, we decided to use the scaled weight to compare connectome with the other methods, which also prioritize interactions in a single condition system. We also include the normalized weight when we use the ‘non-specific’ scoring functions of each method in Supplementary Note 2.

5. OmniPath integrates all the resources together and provides a comprehensive resource to the community. Thus, the quality of the interactions in OmniPath is very important for correct biological interpretations. Does OmniPath include a confidence score or literature to allow users select high confidence interactions?

RIA5. Yes, OmniPath provides information to filter individual interactions by both curation effort, defined as the number of resource-reference pairs which support the interaction, as well as by the literature support, given that the CCC resources integrated in OmniPath provide such information. One can further choose to include only interactions coming from expert-curated resources alone (e.g. CellPhoneDB, ConnectomeDB2020, etc). Moreover, a consensus

between resources can be used to assess the quality of the CCC annotations of proteins, and potentially false interactions can be further filtered by using protein localization and topology data from independent sources.

OmniPath integrates the broadest available knowledge and provides tools to filter the resources and find a balance between quality and coverage. As the reviewer points out, the quality of interactions is crucial to obtain appropriate biological insights. To this end, we have now extended the quality filtering procedures in the OmniPathR R package, and used these to obtain a high quality subset of OmniPath that we used in the manuscript. We further provide a vignette on how to filter OmniPath according to one's preferences both via LIANA and OmniPathR, available at https://saezlab.github.io/liana/articles/liana_custom_op.html and https://saezlab.github.io/OmniPathR/reference/filter_intercell_network.html, respectively.

To make it easier for users to obtain only the curated interactions, we created a new function in our R package OmniPathR which collects only the curated records from LR resources, details available at https://saezlab.github.io/OmniPathR/reference/curated_ligand_receptor_interactions.html.

In addition, the authors categorized the interactions into functional terms such as innate immune pathways. Is such information available in OmniPath? It is useful to have it so that users may select a particular set of interactions, accordingly.

Yes, such information is available for all of the general biological databases from the resource analysis, plus more, accessible via the OmniPathR package. We agree with the reviewer that such information would be of particular use to the user, and thus we have now added a tutorial on how create a customized resource according to intracellular functional term information and how to make further use of it (e.g. over-representation analysis of LIANA predictions), available at https://saezlab.github.io/liana/articles/liana_intracell.html.

6. In Supplementary Table 1, is the number showing in the table the shared interactions or unique ones?

RIA6. We thank the reviewer for noticing this. We have changed the title to reflect that we show the unique interactions alone:

Unique Transmitters, Receivers, and interactions in each resource. We defined unique interactions, receivers and transmitters between the CCC resources if they could be found in only one of the resources.

7. What is the meaning of each row in Figure 4?

RIA7. Each row in Figure 4 represented an interaction which was present (1) or absent (0) for the given method-resource combination. These were then clustered using the 'binary' distances, or Jaccard indices, for each of the method-resource combinations.

Of note, in this version of the manuscript we opted to replace the old Figure 4 with heatmaps of the median Jaccard index (for example **Supp. Figure S13**, see below). We think that this type of heatmaps would be more familiar and easier to interpret for most readers.

Supplementary Figure S13. Heatmap of median Jaccard index between the highest ranked 1,000 interactions per resource-method combination across all datasets. Clustered by Euclidean distance.

8. On page 11, 'CellChat' should be 'CellChatDB'

RIA8. We thank the reviewer for noticing it and we have corrected the text.

Reviewer #2

(Remarks to the Author): Expert in cell-cell communication methods and single-cell RNA-seq

This paper first performed a comparative assessment of known ligand-receptor (LR) resources and then compared all combinations of 15 resources of LR interactions and 6 methods of inferring cell-cell communications (CCC) on a colorectal cancer scRNA-seq dataset. As far as I could understand, the author compared the dependencies and overlap between 15 resources and localization and functional term abundance in CCC resources. On the other hand, the author compared the 500 highest ranked CCC interactions between different combinations of methods and resources. Unfortunately, the comparisons are shallow and biased lacking deep and systematic evaluation as well as informative conclusions on the CCC resources and methods, which makes it hard to see the novelty of this work. It is easy to understand that different combinations are supposed to provide different results as the algorithms underlying methods and the LR pairs underlying CCC resources are different, wherein this work only showed the different results of inferred LR pairs without a comprehensive discussion of the strengths and weakness of each method/resources and a reasonable explanation of the observed disagreement.

We thank the reviewer for their feedback and we acknowledge that the analyses and comparisons were limited in the previous version. In the revised manuscript we have significantly extended the comparisons, including (i) six data sets; (ii) four different benchmarks based on different data modalities (single-cell protein, spatial transcriptomics, cytokine signatures, and robustness), and (iii) an extension of the descriptive comparison to better explain the observed disagreement. We believe that this provides a comprehensive analysis as unbiased as possible and is, to our knowledge, the most complete comparison and evaluation of CCC methods to date. In addition, we provide a new tool to run and integrate in a consistent framework, LIANA, that allows users also to generate consensus predictions.

Specific comments:

1. As mentioned above, the comparisons in Results section are too shallow. More methods (e.g., CytoTalk, NicheNet, SoptSC) are required to be compared.

Although these methods can infer the downstream activated pathways different from the CellChat, CellPhoneDB, SingleCellSignalR, etc., they are the representative CCC methods in this area, which have demonstrated an appealing performance in their benchmarking analysis.

R2A1. We agree with Reviewer 2 that LIANA would strongly benefit by the inclusion of more methods. This is one of the main reasons why we aimed to create a flexible framework that could easily incorporate new approaches in the future.

The manuscript's and LIANA's main objective is to compare and incorporate methods which prioritize ligand-receptor interaction events between pre-annotated cell type populations. As the reviewer notes, some other methods exist but they analyse downstream pathways and can thus not be directly compared to the other methods in LIANA. With this in mind, we thoroughly reviewed the methods suggested by the reviewer, exploring different ways by which they could fit in the manuscript and in LIANA:

- 1) **SoptSC** is a unified framework to infer cell clusters, marker genes, cell pseudo-times, lineages and cell-cell communication events. SoptSC requires an additional input, the set of downstream targets per pathway (with mode of regulation). This precludes comparing it with the methods in LIANA, given that those don't use any other input than the ligand-receptor set of interactions and the single-cell expression data. More importantly, since the SoptSC default approach requires cell clusters/annotations to be predicted using the SoptSC pipeline, we cannot compare SoptSC to the pre-annotated datasets that we used in LIANA.
- 2) **CytoTalk** performs a “de novo” reconstruction of cell-to-cell signaling pathways in two steps: First it creates two intra-cellular signaling pathways, one per cell type, using co-expression measurements. Next, it connects both networks with a user-supplied list of ligand-receptor interactions and employs a prize-collecting Steiner tree algorithm to retrieve the more robust sub-network. To do so, CytoTalk calculates a cross-talk score that is used to guide the sub-network reconstruction. While the whole tool cannot be considered for its inclusion in LIANA, we re-implemented the crosstalk score function, inspired by Cytotalk, and included it as a new method in our analysis (**Table 1**).

This is described in the manuscript:

4.4.7 Crosstalk Scores

Crosstalk scores, inspired by CytoTalk (4), were re-implemented in LIANA. To enable the comparison to the rest of the methods, and in contrast to the crosstalk scores implemented in CytoTalk, we used the inverse of the

non-self-talk scores for autocrine signalling interactions. Non-self-talk scores were originally designed to penalise autocrine signalling. Moreover, our implementation considers complexes, and interactions with transmitters or receivers with preferential expression measures of 0 are also assigned 0.

- 3) **NicheNet**. Differently from the methods included in LIANA, NicheNet requires users to define a hypothesis about the cell-cell communication beforehand, which is then tested using the ligand/target predictions that can be retrieved from its optimized prior knowledge signaling network. That is, NicheNet cannot prioritize connections between clusters in an unsupervised manner, which is arguably the main focus of the methods included in our framework. Moreover, the NicheNet model is independent of the gene expression measurements of ligand/receptor genes in the sender and target cell populations, which limits comparisons with the rest of methods, as they heavily depend on these measurements. Finally, NicheNet does not provide ligand-receptor predictions, but intra-cellular signaling driven by the ligand activity alone. In fact, NicheNet's authors suggest the usage of NicheNet as a complement (<https://tinyurl.com/2p82tzxy>) of the ligand-receptor prediction tools, but not as a substitution of them. We agree with the authors and indeed we consider NicheNet an excellent tool and complementary to LIANA. Hence, we created a dedicated tutorial to exemplify how to apply the NicheNet analysis downstream of LIANA's execution, which is used in this setup as the generator of cell-to-cell communication hypotheses. The detailed vignette is available at https://saezlab.github.io/liana/articles/liana_nichenet.html.

We also implemented a logFC Mean score, inspired by of iTALK:

4.4.4 logFC Mean

The LogFC Mean score implemented in LIANA, was inspired by iTALK (5), which represents the average of one-versus-the-rest logFC changes for the transmitter and receiver. The logFC Mean score uses LIANA's default filtering settings, namely both the transmitter and receiver genes of any interaction evaluated must be expressed in at least 10% of the cells, and it considers the subunit with the minimum expression for complex-containing interactions.

In addition to these methods, we also added a method to aggregate all different methods, and identify the preferentially ranked interactions:

4.4.8 Robust-rank aggregate

A consensus rank is generated across all methods using Robust Rank Aggregation (6). These aggregated ranks can in turn be interpreted as a probabilistic distribution for interactions that are preferentially highly-ranked. The aggregate ranks are built across the universe of all interaction predictions, after independent filtering by each method. By default, missing interactions are imputed as the max ranks.

In summary, we consider that SoptSC, NicheNet, CytoTalk and other methods are very valuable for the analysis of cellular communication, rather complementary than comparable to the pure ligand-receptor CCC methods. We illustrate with NicheNet how they can be used downstream of LIANA, and in the future we will extend LIANA to include further downstream or upstream analyses. We describe this in the Discussion:

Similarly, other analyses such as pathway (7) or transcription factor activities (8,9), as well as other types of cell-communication dedicated methods, including NicheNet (10), CytoTalk (4), and SoptSC (11), could be utilised to provide further confidence in the predicted ligand-receptor interactions.

In addition, more datasets are required to be benchmarked to generate an unbiased comparison.

We agree with the reviewer that more datasets are essential to support one of the main messages in the manuscript. To this end, we extended the number of datasets considered to six. In agreement with our first analysis we saw consistently that both the method and resource have a strong effect on the predictions.

Figure 5. *Overlap (Jaccard index) in the 1,000 highest ranked A) when using the same Method with different Resources (Blue) and B) when using the same Resource with different Methods (Red). Overlap is represented as the pairwise. The dashed lines represent the median when using different resources (red) and methods (blue); the lines overlap for the CMBCs dataset.*

2. Indeed, there is no gold standards in this area to evaluate the CCC resources and methods. However, there are some efficient strategies to evaluate the methods' performance which are widely-used as the authors stated in the manuscript, including a presumed correlation between CCC activity and spatial adjacency, recovering the effect of receptor gene knockouts, robustness to subsampling, agreement with proteomics, simulated scRNA-Seq data, and the agreement among methods. The author also should apply these strategies and the corresponding datasets (e.g., spatial datasets, scRNA-seq datasets with the knockout of ligands/receptors, single-cell multimodal datasets, simulated scRNA-seq datasets) to generate a systematic evaluation.

R2A2. We agree with the reviewer that there are some strategies which could be used to support the performance of the methods, and we thank them for the suggestions.

We considered all suggestions of the Reviewer. Knockout of ligand/receptors would indeed be a meaningful benchmark, but rather for methods that consider changes in gene expression (4) downstream of the receptor instead of those looking solely at ligand-receptors used here. In-silico scRNA-Seq data would be also a sensible option, but existing methods are limited to the simulation of differentially expressed genes (12), and/or few cell types and limited ligand-receptor combinations (13). Considering this, we evaluated this agreement with spatial adjacency (**Results 2.4.1**), cytokine activities (**Results 2.4.2**), and agreement with protein abundance using multi-modal datasets (**Results 2.4.1; Supplementary Figure S23**). Moreover, as the reviewer suggested, we also assessed the robustness of the different methods in terms of both subsampling and reshuffling cluster labels, as well as to diluting resources with ‘false’ interactions (**Supp. Note 2; Supp. Figure S11**).

Figure 6. Odds ratios of **A)** Active cytokines and **B)** colocalized cell types among the highest ranked interaction predictions, across a ranked range between 100 and 10,000. Odds ratios representing the association of preferentially ranked CCC predictions and **A)** cytokine activities and **B)** spatial adjacencies were calculated using Fisher's exact test. Consensus represents the aggregated ranks of all interactions predicted by all the methods. Dashed line is the baseline represented by an odds ratio of 1. The vertical lines represent the truncated ranges of CellChat, CellPhoneDB, and LogFC Mean, arising from their relatively stricter preprocessing steps.

3. In general, it's a descriptive research. A comprehensive discussion of the strengths and weaknesses of each method/resource and a reasonable explanation of the observed disagreement should be provided and more informative and unbiased conclusions should be drawn based on the enlarged benchmarking datasets and comprehensive comparison.

R2A3. We agree with the reviewer that one focus of this work should be around strengths and weaknesses as well as the observed disagreements. To address the latter, we have extended the data sets and data modalities, as mentioned above. For the former, we have now extended the results in 2.3.1 Overlap in interaction predictions, and in more detail Supplementary Note 2, which suggest that the observed differences between the methods largely stem from the different approaches that they use to assign specificity to the predicted interactions. Other reasons for the observed differences include that some methods consider complexes while others do not, and the distinct processing steps used by each of the methods.

Moreover, in the case of the resources we emphasized the importance of manual curation and provided an estimate of curation support for each resource, described in more detail in R3A1.

Reviewer #3

(Remarks to the Author): Expert in cell-cell communication methods, single-cell RNA-seq, and cancer genomics

Summary

The manuscript describes and compares global statistics of the existing resources and tools to infer cell-cell communication (CCC) events from scRNAseq experiments. It represents a comprehensive overview of the existing CCC resources and strategies. It attempts to compare the many published CCC frameworks, a need for the scRNAseq community.

We appreciate that the reviewer values the importance of our work for the scRNAseq community.

Major comments

While the question “how the existing CCC approaches compare?” is extremely relevant, the comparison performed is far from providing guidance in choosing a CCC approach. The manuscript needs additional work, stating and assessing

more useful questions and reducing redundant and nonessential statistics provided.

R3P1. We agree with the reviewer that the paper did not provide enough guidance to choose an approach, and we have thus expanded our analysis with more data sets (2.3.1 Overlap in interaction predictions), a robustness analysis (2.3.2. Robustness to noise in resources and data), and three different benchmarks based on different data modalities (section 2.4. Comparison of predictions with other data modalities).

We also acknowledge the amount of redundant and nonessential statistics provided in our previous text. To this end, we completely rewrote the resource-method comparison (Results 2.3), and removed or replaced the redundant text and statistics with fewer figures that summarise all of the information and clearly convey the message of our work. We also removed the analysis with the HGNC resource and investigations of genesets (defined as the union of receiver and transmitters), as both were largely non-informative. We further strived to reduce the redundant wording and statistics throughout the main text.

Generally, while summary statistics of the resources and tools are informative, the need is more on identifying the approach able to prioritize the CCC key for a tissue, disease or developing process, that can be reproduced and validated in the laboratory. We agree that such an objective is challenging due to few available datasets and the complexity of the questions. Still, we encourage the authors to emphasize that the value of a resource is not a function of the number of interactions reported, but how well these represent experimentally validated interactions with clear role in cell-cell communication.

R2P2. We appreciate that our previous version fell short in providing guidance in relation to the choice of a CCC approach or resource. Although, as the reviewer notes, such a question is challenging to answer with certainty, as mentioned above we extended our analysis to three benchmarks (Results 2.4), which suggested that most methods generally detect the biological signal.

Similarly, the value of the tools should not be measured by the similarity of their results, but by their ability to help the researchers to identify the CCC driving a disease or a differentiation process.

We agree that neither the number of interactions per resource, nor the agreement between methods should be a measure of their quality, and we did not intend to convey this as a message. To make sure that this is clear, we emphasized on the following statements in the discussion:

Some resources are predominantly manually-curated (14–19), while others (5,20–22) are composites which also import non-curated interactions. Thus, this suggests a quality-coverage trade-off, as is commonly the case for biological prior knowledge.

Similarly, we agree that the value of the tools is not measured by the similarity of their results, but rather their ability to generate relevant biological insights, which we now explore with the benchmarks, and we now state this in the discussion:

Overall, our results suggest that despite their relative lack of agreement, the CCC methods are generally able to capture relevant biological signals, and that leveraging information from additional modalities and analyses could help to refine the predictions.

Please find below a series of suggestions and points that need clarification.

1) Resource Uniqueness and Overlap. While the total number of interactions and their overlap can be informative, the more interactions and their uniqueness is not a measure of quality. Especially when comparing a) resources that uniquely contain their own “expert curated” interactions (such Ramilowski2015/NATMI, CellphoneDB or HPMR) to b) other resources that import interactions from (a) or include interactions that have not been curated by experts. We suggest the authors distinguish between expert curated and non-expert curated interactions. A curated interaction is that with a supported role in cell-cell communication (i.e. induces a response in the interacting cells or has a structural role - e.g. adhesion molecules). An interaction between a secreted protein and a membrane bounded protein identified by co-immunoprecipitation or yeast-two hybrid is not a curated interaction as their role in cell-cell communication is not experimentally supported.

R3A1. We agree with the Reviewer that the amount of unique interactions is not a measure of quality. Instead, it can be interpreted either that the resource brings largely curated and ‘novel’ knowledge, or can be a sign that the resource possibly contains false interactions. In case of manually-curated resources, we tend to expect the former and the latter for high-throughput, text mined or in silico inferred data.

We thus share the Reviewer’s interest about distinguishing the curated interactions as these are generally deemed to be higher quality, and expert curation is a valuable, unique effort from the resource authors. Hence we examined each resource whether it contains original curation and if it's possible to distinguish the curated records.

To highlight the manual curation efforts, we included this information in two new columns in Supp. Table 2.

One of these columns shows whether the resource contains original curated interactions (Original curation † § in Supplementary Table 2):

† Does the resource contain original expert curated data? Here *expert curated* (synonyms: manually- or literature curated) means only the interactions curated in the context of cell-cell communication. We assessed the curation contents based on the publications, webpages, software manuals and data files.

We also attempted to estimate the percentages of these (See below); both assessed based on the publications, webpages and data. We decided to not include this, since for most of the resources, due to ambiguity of the data or insufficient documentation, it’s not completely certain which records come from curation and which ones are imported from other curated resources (e.g. Baccin and LRdb):

Resource	Own curation
Baccin2019	82%
CellCall	0%
CellChatDB	30%
Cellinker	26%
CellPhoneDB	33%
CellTalkDB	91%
ConnectomeDB2020	100%
EMBRACE(a)	0%
Guide2Pharma	44%
HPMR	92%
ICELNET	100%
iTALK	0%
Kirouac2010	100%
LRdb	62%
Ramilowski2015	14%
talklr	11%
scConnect	61%

We also collected the original curated interactions from all resources and in another column we show the overlap of each resource with this curated set (Overlap with curated set ♦ in Supplementary Table 2):

♦ We collected the curated interactions from all resources which contained any. Then for each resource we measured the overlap against the curated set. Many resources contribute to the curated set, or integrate other resources which are part of it, resulting in a bias in the overlaps.

We also show the resources with original curation effort on Figure 2.

Figure 2. Dependencies and overlap between CCC resources. The lineages of CCC interaction database knowledge. General biological knowledge databases (blue), CCC-dedicated resources used in this work (magenta), manual literature curation effort (yellow), additional resources included in iTALK (cyan), and OmniPath (green). Arrows show the data transfers between resources.  indicates the manually-curation of resources, defined by explicitly mentioning that these resources are ‘manually’ or ‘expert’ curated.  indicates that the resource was included in the analyses presented here.

Finally, to make it easier for users to obtain only the curated interactions, we created a new function in our R package OmnipathR which collects only the curated records from LR resources (https://saezlab.github.io/OmnipathR/reference/curated_ligand_receptor_interactions.html).

We also explicitly discuss the reviewers' point in the discussion:

Some resources are predominantly manually-curated (14–19), while others (5,20–22) are composites which also import non-curated interactions. Thus, this suggests a quality-coverage trade-off, as is commonly the case for biological prior knowledge. Of note, the literature-support reported by different authors for the same resources do not always agree (15,23), suggesting different interpretations of what defines a curated interaction.

Other pertinent questions are:

- Out of the total number of candidate CCC interactions reported by all these resources: how many interactions are supported by experts or have a proven role in cell-cell communication (or alternatively how many are potentially novel candidates)? This will help the reader to understand the size of the bona fide or candidate CCC.
- What resources are contributing to each type of interaction? This can help the reader to understand the risks (or the likelihood of finding novel CCC) in choosing a resource.
- What % of curated interactions are borrowed from other resources? Which resource incorporates the largest number of non-borrowed expert-curated CCC?

To address these questions, as described above (R3A1), we added assessments of expert curated contents to **Supp. Table 2**, and thus information suggested by the Reviewer is covered by the **Overlap with curated set** and **Original curation** columns.

Then we looked at the overlap between the curated interactions among the resources (**Supp. Figure 2 panel D**):

Supplementary Figure S2. D) Curated interactions [...] present in each resource when taken from the rest of the resources. Note these plots are asymmetric and represent the % of interactions from the resources on the X axis found in each resource on the Y axis.

These results largely resembled what we observed in Supp. Figure 2A, namely that many of the subsequently published resources borrow curated interactions from Ramilowski, among others.

We also highlight the most frequently imported curated resources in the text:

Moreover, interactions from the Guide to Pharmacology (24), CellPhoneDB (14), HMPR (25), and in particular Ramilowski (FANTOM5) (26), which are manually curated, were commonly incorporated into subsequently published resources (Figure 2A; Supp. table 2)

This information can be included as edge width in figure 2a, and will acknowledge the contribution of each resource to the CCC census.

In Figure 2, the edge widths wouldn't be readable as the figure is already quite busy and we thus included only a qualitative attribute about the original curation content of each CCC resource, and provide the complete information in Supplementary Figure S2.

- The amount of unique CCC reported by Omnipath and CellChatDB is worrying. What is the evidence for these interactions? What are the proteins that more frequently appear in the Omnipath and CellChatDB unique categories? Can the authors provide these?

We thank the reviewer for noticing this and for their comment. Indeed, after submission, we found a processing issue in both resources. In the case of OmniPath, this was due to an erroneous processing of certain records from the SIGNOR database. For CellChatDB it was an issue associated with the inappropriately labelling of mediator proteins. Both these issues have now been resolved and now the number of unique CCC is **8.45%** and **20.76%** instead of **50.29%** and **45.70%** for Omnipath and CellChat, respectively.

2) The classification of proteins as receivers or transmitters is confusing. Except for CCC involving secreted ligands, the remaining interactions likely have a bi-directional effect on the interacting cells.

R3A2. We agree with the reviewer that the bi-directional effect is a very valid point and this was also a topic of long discussions among ourselves. The reason why we chose the receiver-transmitter was to distance ourselves from the commonly perceived (secreted) ligand and receptor interaction, which itself suggests a one-directional effect. We defined our current terminology in Turei et al. 2021 (27), Dataset EV10 (accessible at <https://tinyurl.com/4se5x9en>). Many of the proteins are both transmitters and receivers, hence bi-directional connections are possible. That said, we are open to other nomenclature that might be more appropriate.

Are adhesion molecules, such as integrins, reported as receivers or transmitters? Are these ignored or counted twice?

In the non-filtered OmniPath resource, potentially bi-directional interactions, such as adhesion molecules or integrins, are labelled as both transmitter and receiver. For details of the classification see Turei et al., 2021 (27), Dataset EV10 (available at <https://tinyurl.com/4se5x9en>).

3) For resources considering complexes, how are interactions quantified and compared to the resources ignoring the complex structure? Are the complexes decomposed in pairs (for example is simple(A) <-> complex(B1+B2) decomposed as A-B1 and A-B2 rendering two pairs of interactions) or are they kept as a unit (A-B1/B2)? This is important because for two resources with the exact same information, the one ignoring the complex structure will systematically have more interactions.

R3A3. This is indeed an important point. We dissociated complexes into subunits for all resource comparisons, as this enables a direct comparison. This

was also the case when considering each of the general knowledge databases. To make it clearer we included this in the main text:

These latter [complex-containing] resources [...] include protein complexes, which were dissociated and treated as distinct protein subunits in our resource analyses.

And also in the Methods:

All complex-containing resources were dissociated into individual subunits for the resource-focused analyses presented in this work.

Each of the aforementioned general functional annotation databases was obtained via OmniPath and their protein complexes, if present, were also dissociated.

4) We don't find that the section "2.2.1 Subcellular Localisation" provides critical information to choose a resource. This section can be shortened. It is expected that the majority of resources will be biased towards secreted proteins and membrane bound receptors as these are the most obvious CCC candidates for both manual curation and in silico predictions. Pharmacology, for obvious reasons, will be biased towards secreted proteins.

R3A4. We agree with the reviewer that the information provided by **2.2.1 Subcellular Localisation** was not a key message of our manuscript. Since the results can be informative to certain readers, e.g. if one wishes to use a resource enriched for direct-contact interactions, we kept the analysis, but moved it to **Supp. Note 1**.

5) Also, not sure what the analysis with "Functional cancer cell states from CancerSEA" adds here. It is hard to predict what impact "DNA damage states were over-represented in LRdb " will have, as the link between cell-cell interactions and DNA damage is ambiguous.

R3A5. We agree that DNA damage states were not very informative. We have thus redone our analysis with CancerSEA to instead consider the interactions. We obtained results associated with states such as interaction distributions associated with **inflammation** and **proliferation**, which we consider of high importance for the prediction of interactions associated with signalling in cancer tissues (**Figure 4C**).

A more useful information would be to know the **bias towards particular tissues/diseases**. For example, if I am studying the brain, which is the resource covering larger % of brain-specific proteins (and the % of curated CCC)? This is also relevant for immune studies. Immune cells are better characterized because of tissue accessibility. Resources that uniquely incorporate manually curated CCC may likely be biased towards immune CCC. One could use GTEx or human atlases to identify tissue or lineage-specific genes.

We agree with the reviewer that examining the relative abundance of tissue markers and disease. To this end, we extended our work to include organ (Figure 4D) and cell type markers (Supp. Figure S8) from the Human Protein Atlas, as well as literature-supported disease markers from DisGeNet (Supp. Figure S9). We have now a new dedicated section: **2.2.2 Tissue and Disease Marker Enrichment**.

Figure 4. CCC resources distributions in terms of number of interactions (A) and relative abundance (B) matched to the SignaLink database. C) Interactions categorised by CancerSEA cancer cell states, and D) Human Protein Atlas organ markers. Differentially represented ($\log_2(\text{Odds ratio}) > 1$) categories were marked according to FDR-corrected p -values ≤ 0.05 (⚡), 0.01 (*), and 0.001 (*).

6) Regarding the “Agreement in CCC predictions”, the authors “looked at the overlap between the 500 highest ranked interactions as predicted by each method”. Can Table 1 be extended to include what measurement from each tool

they use as a ranking score? Are these scores a measure of specificity (cluster-enriched), mRNA abundance (avg. expression), etc?

R3A6. We thank the reviewer for the suggestion to clarify the different scoring functions used throughout the manuscript. Since we could not fit this information in **Table 1**, we have added it in a new **Supp. Table 3**.

Supplementary table 3. *Description of the different scoring settings used to compare the methods presented in this work.*

	Composite#		Non-composite	Non-Specific
Method	Used throughout the main text		Specificity alone for CellChat and CellPhoneDB	Scores which do not incorporate cell-pair specificity in interaction predictions
CellChat	Probability	(Filtered by p-value < 0.05)	p-values alone	Probability
CellPhoneDBv2	Truncated Mean	(Filtered by p-value < 0.05)	p-values alone	Truncated Mean
Connectome	weight_scale		weight_scale	weight_norm
Crosstalk Scores	Crosstalk score		Crosstalk score	-
logFC Mean	logFC Mean		logFC Mean	-
NATMI	Specificity-based edge weight		Specificity-based edge weight	Mean-expression edge weight
SingleCellSignalR	LRscore		LRscore	LRscore
# Unless explicitly mentioned, we used the composite method settings.				

7) The fact that the results depend on the method and not on the resource is unexpected. Can the authors elaborate more on the reasons why this can be the case? For example, are the methods sensitive to the size of the clusters?

R3A7. From this comment of the Reviewer, we realise that we should have explained more clearly that the results depend both on the method and the resource. In this new version of the manuscript, we have revised the text to clarify this. Also, we now dedicate a section to the reasons why we observe the differences between the methods (**Supplementary Note 2**). We have also updated the relevant section: **2.3 Using LIANA to systematically compare CCC predictions**.

and also added this sentence from the discussion:

Our results suggest that both the method and resource can considerably impact CCC inference predictions [...]

Do all the methods normalize the input data so that measurements are comparable within the dataset?

All methods were used with the log-normalized counts, or if recommended using conversions provided by the authors. We added the following sentence to clarify this to the reader:

For the method-resource comparisons and evaluations, we used Seurat (28,29) objects which were converted to the appropriate data format when calling each method. Whenever available, we used the recommended conversion method or wrapper for each method. Log-transformed counts were used when this was not done internally by the method.

8) The “interaction overlap” between methods, although useful to highlight the differences between approaches, does not inform on their ability to retrieve relevant biology. In the absence of a gold standard, it may be useful to quantify the enrichment in cluster-specific genes in the 500 highest ranked interactions predicted by each method. This can be done with classical differential gene expression (DGE) analysis or alternative methods looking for specificity, such as TF-IDF. Cluster-specific genes will be a constant in each annotated dataset and could be a good reference to compare methods under the same resource. A CCC tool that misses most of these genes (with a CCC role) may not be that appropriate. It should be noted that CellChat and iTALK already rely on DEGs.

R3A8. We appreciate the Reviewer's suggestion, that we took into account when developing our new benchmarks with the additional data modalities in section Association between CCC predictions and Cytokine Expression Signatures (Results 2.4.1), and in particular Agreement with Receptor Protein specificity (Supp. Note 3). In the latter, we considered the cluster-specific protein abundance to evaluate the accordance of the CCCs and the expression of the corresponding receptors.

9) The biological value of annotating the complexes should be highlighted in the discussion as ignoring such nature will generate false positives, no matter the tool used. For example, if a receptor is a heterodimer and one unit is not expressed, the CCC is just not possible. Instead, the authors mention “resources, such as Baccin, CellChatDB, CellPhoneDB, and ICELLNET, take protein complexes into account. This largely complicates the conversion of the resources”. Again, the focus should be on the quality of the predictions and not

on the comparability of the resources, especially when simplification induces false signals.

R3A9. We agree that we have overemphasized on the difficulties, without appropriately highlighting the reduction of false positive results. To address this we added the following sentence in the Discussion:

Some of the methods such as CellChat (7), CellPhoneDB (14), and others (17,18,22), go a step further by considering heteromeric complexes, which has been shown to reduce false positive predictions (7,14). CellChat also accounts for interaction mediator proteins (7).

Resources can be compared if these are decomposed into the same units.

We agree with the reviewer, and we have indeed done so (see **R3A3**).

10) The authors use the word “activity” several times in the text. Activity implies signal transmission and is a concept that only methods accounting for downstream signaling actually explore (such NicheNet). What do the authors mean by “CD4+ T and CD8+ T cells ... were assigned a CCC communication activity.” How are the authors quantifying CCC activity? In the same sense, what does the sentence “actively communicating cell clusters” mean? A single CCC interaction may be enough to have an “active” communication between two cells (for example a strong ligand inducing cell death). If the authors use activity as a function of the “number of CCC reported between two cell types”, we strongly suggest they rephrase the text to avoid confusion.

R3A10. We acknowledge that cell type “activity” was inaccurately used in the first version of the manuscript for the relative proportions of interactions assigned to different cell types. In this version, we have completely refrained from using “activity” in this context, and we instead referred to the same analysis as interaction frequency per cell type in **Supplementary Note 2**.

Minor comments

- The text should be shortened for effectiveness. Most of the statistics described do not add but distract the reader from the main message.

Agreed. Following the Reviewer's suggestion, we focused on providing appropriate figures, which enabled us to refrain from the use of redundant

statistics, thus shortening our text and appropriately conveying the main message. In particular, major changes were done to **Results 2.3**.

- CellphoneDB is a tool as well (Figure 1). Actually, CellphoneDB is among the leading tools in the field (with ~300 citations), while Squidpy independent implementation of this tool has not been peer reviewed yet. CellphoneDB's original tool should be included in the comparison.

Indeed, CellPhoneDB and the creators of its algorithm should have been accredited. To this end, we have changed **Figure 1** to show the name of the original algorithm and implemented it in LIANA. As the reviewers suggest, Squidpy is an independent framework with a distinct purpose.

- Why are the authors not including the tool NicheNet? We agree that NicheNet is not directly comparable as it accounts for downstream signalling, but at least should be mentioned and listed as a CCC method.

This question, which is a valid one, was already discussed in detail in point **R2A2**. To summarize, NicheNet is complementary to the methods discussed here, and as the reviewer says not directly comparable. We now mention it in the table and text, and provide a vignette to run it downstream of LIANA:

https://saezlab.github.io/liana/articles/liana_nichenet.html

- Sentence “By focusing on the cluster-specific interactions in the dataset, these methods report the most specifically-interacting cell types”. The methods do not report interacting cell types but the interacting proteins between two cell types. What is tested for specificity is the protein/cell type combo. An alternative could be “By focusing on the cluster-specific interactions in the dataset, these methods report the interacting proteins used specifically/exclusively by a pair of cell types”.

The related text was corrected.

- In the sentence “Such an assumption inherently suggests that gene expression is informative of the activity of transmitters and receivers. However, gene expression provided by scRNA-Seq is typically limited to protein coding genes and the cells within the dataset”. “However” should be replaced by “also, additionally” as sentences are independent/additive.

The related text was corrected.

Bibliography

1. Gregory AP, Dendrou CA, Attfield KE, Haghikia A, Xifara DK, Butter F, et al. TNF receptor 1 genetic risk mirrors outcome of anti-TNF therapy in multiple sclerosis. *Nature*. 2012 Aug 23;488(7412):508–11.
2. Iorio F, Bernardo-Faura M, Gobbi A, Cokelaer T, Jurman G, Saez-Rodriguez J. Efficient randomization of biological networks while preserving functional characterization of individual nodes. *BMC Bioinformatics*. 2016 Dec 20;17(1):542.
3. Raredon MSB, Yang J, Garritano J, Wang M, Kushnir D, Schupp JC, et al. *Connectome*: computation and visualization of cell-cell signaling topologies in single-cell systems data. *BioRxiv*. 2021 Jan 21;
4. Hu Y, Peng T, Gao L, Tan K. CytoTalk: De novo construction of signal transduction networks using single-cell transcriptomic data. *Sci Adv*. 2021 Apr 14;7(16).
5. Wang Y, Wang R, Zhang S, Song S, Jiang C, Han G, et al. iTALK: an R Package to Characterize and Illustrate Intercellular Communication. *BioRxiv*. 2019 Jan 4;
6. Kolde R, Laur S, Adler P, Vilo J. Robust rank aggregation for gene list integration and meta-analysis. *Bioinformatics*. 2012 Feb 15;28(4):573–80.
7. Jin S, Guerrero-Juarez CF, Zhang L, Chang I, Ramos R, Kuan C-H, et al. Inference and analysis of cell-cell communication using CellChat. *Nat Commun*. 2021 Feb 17;12(1):1088.
8. Jakobsson JET, Spjuth O, Lagerström MC. scConnect: a method for exploratory analysis of cell-cell communication based on single cell RNA sequencing data. *Bioinformatics*. 2021 May 11;
9. Jung S, Singh K, del Sol A. FunRes: resolving tissue-specific functional cell states based on a cell–cell communication network model. *Brief Bioinformatics*. 2021 Jul 20;22(4).
10. Browaeys R, Saelens W, Saeys Y. NicheNet: modeling intercellular communication by linking ligands to target genes. *Nat Methods*. 2020 Feb;17(2):159–62.
11. Wang S, Karikomi M, MacLean AL, Nie Q. Cell lineage and communication network inference via optimization for single-cell transcriptomics. *Nucleic Acids Res*. 2019 Jun 20;47(11):e66–e66.
12. Tsuyuzaki K, Ishii M, Nikaido I. Uncovering hypergraphs of cell-cell interaction from single cell RNA-sequencing data. *BioRxiv*. 2019 Mar 4;
13. Tanevski J, Ramirez Flores RO, Gabor A, Schapiro D, Saez-Rodriguez J. Explainable multi-view framework for dissecting inter-cellular signaling from highly multiplexed spatial data. *BioRxiv*. 2020 May 10;

14. Efremova M, Vento-Tormo M, Teichmann SA, Vento-Tormo R. CellPhoneDB: inferring cell-cell communication from combined expression of multi-subunit ligand-receptor complexes. *Nat Protoc.* 2020 Apr;15(4):1484–506.
15. Shao X, Liao J, Li C, Lu X, Cheng J, Fan X. CellTalkDB: a manually curated database of ligand-receptor interactions in humans and mice. *Brief Bioinformatics.* 2021 Jul 20;22(4).
16. Hou R, Denisenko E, Ong HT, Ramilowski JA, Forrest ARR. Predicting cell-to-cell communication networks using NATMI. *Nat Commun.* 2020 Oct 6;11(1):5011.
17. Zhang Y, Liu T, Wang J, Zou B, Li L, Yao L, et al. Cellinker: a platform of ligand-receptor interactions for intercellular communication analysis. *Bioinformatics.* 2021 Jan 20;
18. Noël F, Massenet-Regad L, Carmi-Levy I, Cappuccio A, Grandclaude M, Trichot C, et al. Dissection of intercellular communication using the transcriptome-based framework ICELLNET. *Nat Commun.* 2021 Feb 17;12(1):1089.
19. Jin S, Guerrero-Juarez CF, Zhang L, Chang I, Myung P, Plikus MV, et al. Inference and analysis of cell-cell communication using CellChat. *BioRxiv.* 2020 Jul 22;
20. Cabello-Aguilar S, Alame M, Kon-Sun-Tack F, Fau C, Lacroix M, Colinge J. SingleCellSignalR: inference of intercellular networks from single-cell transcriptomics. *Nucleic Acids Res.* 2020 Jun 4;48(10):e55.
21. Sheikh BN, Bondareva O, Guhathakurta S, Tsang TH, Sikora K, Aizarani N, et al. Systematic Identification of Cell-Cell Communication Networks in the Developing Brain. *iScience.* 2019 Nov 22;21:273–87.
22. Baccin C, Al-Sabah J, Velten L, Helbling PM, Grünschläger F, Hernández-Malmierca P, et al. Combined single-cell and spatial transcriptomics reveal the molecular, cellular and spatial bone marrow niche organization. *Nat Cell Biol.* 2020 Jan;22(1):38–48.
23. Zhang Y, Liu T, Hu X, Wang M, Wang J, Zou B, et al. CellCall: integrating paired ligand-receptor and transcription factor activities for cell-cell communication. *Nucleic Acids Res.* 2021 Sep 7;49(15):8520–34.
24. Harding SD, Sharman JL, Faccenda E, Southan C, Pawson AJ, Ireland S, et al. The IUPHAR/BPS Guide to PHARMACOLOGY in 2018: updates and expansion to encompass the new guide to IMMUNOPHARMACOLOGY. *Nucleic Acids Res.* 2018 Jan 4;46(D1):D1091–106.
25. Ben-Shlomo I, Yu Hsu S, Rauch R, Kowalski HW, Hsueh AJW. Signaling receptome: a genomic and evolutionary perspective of plasma membrane receptors involved in signal transduction. *Sci STKE.* 2003 Jun

17;2003(187):RE9.

26. Ramilowski JA, Goldberg T, Harshbarger J, Kloppmann E, Lizio M, Satagopam VP, et al. A draft network of ligand-receptor-mediated multicellular signalling in human. *Nat Commun.* 2015 Jul 22;6:7866.
27. Türei D, Valdeolivas A, Gul L, Palacio-Escat N, Klein M, Ivanova O, et al. Integrated intra- and intercellular signaling knowledge for multicellular omics analysis. *Mol Syst Biol.* 2021;17(3):e9923.
28. Satija R, Farrell JA, Gennert D, Schier AF, Regev A. Spatial reconstruction of single-cell gene expression data. *Nat Biotechnol.* 2015 May;33(5):495–502.
29. Butler A, Hoffman P, Smibert P, Papalexi E, Satija R. Integrating single-cell transcriptomic data across different conditions, technologies, and species. *Nat Biotechnol.* 2018 Jun;36(5):411–20.

REVIEWER COMMENTS

Reviewer #1 (Remarks to the Author):

The authors have well addressed most of my

concerns. In response to Comment #2, the authors used Jaccard indices to compute the consistency when introducing false interactions.

However, this metric is not able to provide information on the ratio of recalled interactions as well as the percentage of false positive interactions introduced by the false resources. It may be more informative to compute both the true positive rate and false positive rate.

Reviewer #2 (Remarks to the Author):

In general, the authors have addressed the comments from me and other reviewers.

Reviewer #3 (Remarks to the Author):

The authors have improved the quality of their assessment and the manuscript as a whole. The work behind LIANA is remarkable. Overall, the authors addressed most of the points raised in the first round of revision, however, we still find some conceptual issues (e.g. "protein complexes", "tissue markers", "cell state"). Importantly, we also noticed some issues with the new benchmarks ("Comparison of predictions with other data modalities" section) that should be clarified before the manuscript can be accepted for publication.

Comments regarding the authors' responses (in the same order).

- R3A1. Please check the end of the Supp. Table 2 as there is some strikethrough text.

- R3A5. In Figure 4d, most organs are not represented, how are the authors defining tissue-associated proteins here? In the introduction, the authors mention that they “explored ... whether certain resources are biased toward ‘tissue markers’”, did the authors use “tissue markers” only for Figure 4d? Note the difference between tissue markers and tissue-enriched proteins (i.e., “specificity” term according to HPA). We would expect the authors to use tissue-enriched proteins, not tissue markers.

- R3A5. Figure 4a may benefit from accounting “% interactions” instead of “total interactions” in the Y-axis, since the size of the resources already discussed.

- R3A9. We recommend the authors to highlight further the relevance of accounting for protein complexes in cell-cell signalling, and elaborate on why they reduce false positives. A complex is not functional if a subunit is missing. Also, different heterodimeric combinations may render different responses and activate different downstream signals. This is key for interpretation and validation. Following this reasoning, we recommend the authors to amend their sentence in the introduction accordingly “Some of these resources also provide additional details for the interactions such as information about protein complexes, subcellular localisation”. Protein complexes are not “additional details” but the functional units of cell-cell communication.

Comments on the novel analysis.

- “Comparison of predictions with other data modalities” using cytokine activities. The authors employ a collection of 43 cytokine activity signatures to test whether top ranked interactions are positively enriched in cytokines. To this extent, the authors use a multivariate linear methodology to obtain a cytokine score on the pseudobulk level. Cytokines with positive scores and significance are then employed as 'ground truth'. We recommend the authors to provide more details on the multiple steps involved in this analysis. For example:

How is the cytokine score generated? We assume this score is generated for pseudobulk cells within a cell-type, but should be clarified.

Did the authors run CCC on the pseudobulk cell profiles as well?

It is not evident which cells of the dataset are being employed to perform the CCC analysis. Are the interactions ranked across all pairs of interacting cells?

How are odds ratios computed? A cytokine receptor can appear in several top ranked interactions, how are these “duplicates” treated?

As this analysis is treated like a classification problem, it could be of interest providing the results based on ROC/AUC statistics rather than on a Fisher's exact test. This could provide an approximated idea of the trade-off (true positives vs false positives i.e. active/inactive cytokines) yielded by the different CCC methodologies.

More globally, in the in vivo datasets, it should be expected that a cell is influenced by multiple cytokines. However, as far as we understand, combined response signatures are not taken into account. We recommend the authors to clearly highlight the limitations of using such signatures as ground truth.

- “Comparison of predictions with other data modalities” using spatial data. We find a conceptual misunderstanding in the use of spatial colocalisation to benchmark CCC methods. The “assumption that their highest ranked interactions should be positively associated in interactions between pairs of adjacent cell types” is misleading. Molecules involved in cell-cell interactions usually do not have a 1:1 specificity, however multiple combinations of L/R or interacting adhesion molecules exist. Just to mention a few examples, collagens/integrins or TGFb/Notch/WNT signalling molecules may work through different combinations. Let’s consider the following scenario:

(i) Interactions the resource: JAG1-NOTCH1, JAG1-NOTCH2, JAG2-NOTCH1, JAG2-NOTCH2. Note that there are more Notch receptors (e.g. NOTCH3), but let’s consider only these for simplicity.

(ii) Transcriptomic profile: cell1 overexpressing JAG1+, cell2 overexpressing NOTCH1+, cell3 overexpressing JAG2+, cell4 overexpressing NOTCH2+. Assume overexpression occurs at similar levels.

(iii) Colocalization: cell1 and cell2 colocalise, but not cell3 or cell4.

Under these conditions, one can not expect a CCC method to rank higher cell1-cell2 over cell1-cell4 or cell3-cell2 or cell3-cell4 without prior knowledge on neighbouring cells. Accordingly, we do not consider that spatial data represents a gold standard dataset for comparing CCC methods. Instead, it can be very valuable to refine CCC predictions. Instead, information on pairs of adjacent cell types can be used to discard interactions that are infeasible due to physical distance between the cells.

- The low performance of CellChat, CellPhoneDB and SingleCellSignaR, in at least one of the datasets employed, might be due to the fact that the composite scores employed by these tools do not allow for a direct comparison between scores generated for different ligand-receptor combinations. The scores of these tools depend on the input value of the ligands and receptors, while other methods rely on scaled scoring methodologies allowing for a more direct comparison between pairs of ligands and receptors. Can the authors elaborate on this?

- Some result sections would benefit from highlighting their biological implications. For example “the similarity among the resources was generally higher when considering transmitters and receivers, rather than the interaction themselves” one could add, “meaning that different resources account for different interactions for the same proteins”.

- In page 11, what is a “functional cell state”? How can a “cell state” be underrepresented in a CCC database?

Legend:

Response

Reference to Main text

Suggested text changes

REVIEWER COMMENTS

Reviewer #1 (Remarks to the Author):

The authors have well addressed most of my concerns.

We are glad that our work has addressed most of the reviewer's concerns.

In response to Comment #2, the authors used Jaccard indices to compute the consistency when introducing false interactions.

However, this metric is not able to provide information on the ratio of recalled interactions as well as the percentage of false positive interactions introduced by the false resources. It may be more informative to compute both the true positive rate and false positive rate.

We thank the reviewer for the suggestion. We have now computed both the true positive rate and false positive rate. As can be seen from the figures below (Response Figure 1), in our setting the False Positive Rate would not be very informative, as the negative class vastly outnumbers the positive class). For this reason, in general we focused on the ability of methods to correctly identify positives in all settings. We accordingly include the true positive rate figure in the manuscript, replacing the Jaccard Indices.

Response Figure 1. Robustness to A) Cell type subsampling, B) Reshuffling of cell types labels C) Selective, and D) Non-selective replacement of interactions with putatively false ones. The non-modified predictions (0% modifications) from each method were used to calculate the False Negative Rate.

C)

D)

Supplementary Figure S13. Robustness to A) Cell type subsampling, B) Reshuffling of cell types labels C) Selective, and D) Non-selective replacement of interactions with putatively false ones. The non-modified predictions (0% modifications) from each method were used to calculate the True Positive Rate.

Reviewer #2 (Remarks to the Author):

In general, the authors have addressed the comments from me and other reviewers.

We thank the reviewer for acknowledging our effort in addressing the previously raised comments.

Reviewer #3 (Remarks to the Author):

The authors have improved the quality of their assessment and the manuscript as a whole. The work behind LIANA is remarkable.

We appreciate that the reviewer recognizes the improved quality of our work.

Overall, the authors addressed most of the points raised in the first round of revision, however, we still find some conceptual issues (e.g. “protein complexes”, “tissue markers”, “cell state”). Importantly, we also noticed some issues with the new benchmarks (“Comparison of predictions with other data modalities” section) that should be clarified before the manuscript can be accepted for publication.

We are glad that we could address most of the reviewer’s points. We further thank the reviewer for thoroughly checking our manuscript again and for their clarification suggestions.

We have now clarified and elaborated on our choices and assumptions in the “Agreements with other modalities” sections. We also extended our Methods section, accordingly adjusted the text and figures as suggested, and further described the biological relevance of our results.

Comments regarding the authors' responses (in the same order).

- R3A1. Please check the end of the Supp. Table 2 as there is some strikethrough text.

P1. We thank the reviewer for noticing and we have now removed the strikethrough text.

- R3A5. In Figure 4d, most organs are not represented, how are the authors defining tissue-associated proteins here?

P2. We defined the enriched proteins as those with a medium/high level of expression, excluding any proteins with missing or ‘uncertain’ reliability scores according to the HPA database.

In the introduction, the authors mention that they “explored ... whether certain resources are biased toward ‘tissue markers’ ”, did the authors use “tissue markers” only for Figure 4d?

We separately used ‘organ’- and ‘tissue’-enriched proteins from the 15 largest categories in **Figure 4D** as well as in **Supp. Figure S7** and **Supp. Figure S9**. The 15 largest categories were defined as those with the highest overlap with the collective of all CCC resources. Note that we refer collectively to the analyses in the aforementioned figures as ‘tissue’-enriched proteins, as this is how they are represented in HPA, yet it is worth noting that the cut between what is defined as ‘organ’ and ‘tissue’ in HPA can be vague. For example, one could see tissues in their ‘organ’ category (e.g. Adipose tissue), while their ‘tissue’ category is largely synonymous with cell type, as the data in HPA was primarily based on immunohistochemistry at the cell-type level resolution.

Note the difference between tissue markers and tissue-enriched proteins (i.e., “specificity” term according to HPA). We would expect the authors to use tissue-enriched proteins, not tissue markers.

Indeed we used the term ‘markers’ incorrectly, and we now call them ‘enriched proteins’.

Furthermore, we have now updated our figures to the most recent version of the HPA (v21.0), and carried out the tissue-enriched protein analysis by grouping by both organ and tissue, rather than by tissue alone (**Supp. Figures S9-10**).

Finally, we generated results for both the organ- and tissue-enriched protein analysis considering the largest 50 categories:

Supplementary Figure S8. Interaction enrichment scores were calculated for the largest 50 categories matched to organ-enriched proteins from the HPA database. Differentially represented ($\log_2(\text{Odds ratio}) > 1$) categories were marked according to FDR-corrected p -values ≤ 0.05 (\clubsuit), 0.01 (\ast), and 0.001 (\spadesuit).

Supplementary Figure S10. Interaction enrichment scores were calculated for the largest 50 categories matched to tissue-enriched proteins from the HPA database. Differentially represented ($\log_2(\text{Odds ratio}) > 1$) categories were marked according to FDR-corrected p -values ≤ 0.05 (✦), 0.01 (✧), and 0.001 (✨).

- R3A5. Figure 4a may benefit from accounting “% interactions” instead of “total interactions” in the Y-axis, since the size of the resources already discussed.

P3. As suggested, we have now swapped the places of **Figure 4A** with **Supp. Fig 3D**, which shows the % of interactions instead.

- R3A9. We recommend the authors to highlight further the relevance of accounting for protein complexes in cell-cell signalling, and elaborate on why they reduce false positives. A complex is not functional if a subunit is missing. Also, different heterodimeric combinations may render different responses and activate different downstream signals. This is key for interpretation and validation. Following this reasoning, we recommend the authors to amend their sentence in the introduction accordingly “Some of these resources also provide additional details for the interactions such as information about protein complexes, subcellular localisation”. Protein complexes are not “additional details” but the functional units of cell-cell communication.

P4. We agree with the reviewer regarding the functional value of complexes. To this end, according to the reviewer’s comment, we have now amended our introduction to better reflect the functional relevance of protein complexes:

Some of these resources also provide additional details for the interactions such as information about ~~protein complexes~~, subcellular localisation^{1,2}, classification into signalling pathways and categories^{1,3} (**Supp. table 1**). Notably, some resources¹⁻⁵ (**Supp. table 1**), and consequently their corresponding methods, focus on protein complexes as the functional units of CCC, which are crucial for the coordination of signalling as different subunit combinations may induce distinct responses⁴.

We also extended our discussion accordingly:

Some of the methods such as CellChat¹, CellPhoneDB⁴, and others^{3,5,6}, go a step further by considering heteromeric complexes. Ensuring that all units of a protein complex are expressed to consider it as a valid cell-cell interaction candidate has been shown to reduce false positive predictions, and can thus impact significantly downstream interpretation and validation^{1,4}.

Comments on the novel analysis.

- “Comparison of predictions with other data modalities” using cytokine activities. The authors employ a collection of 43 cytokine activity signatures to test whether top ranked interactions are positively enriched in cytokines. To this extent, the authors use a multivariate linear methodology to obtain a cytokine score on the pseudobulk level. Cytokines with positive scores and significance are then employed as 'ground truth'. We recommend the authors to provide more details on the multiple steps involved in this analysis. For example:

How is the cytokine score generated? We assume this score is generated for pseudobulk cells within a cell-type, but should be clarified.

Did the authors run CCC on the pseudobulk cell profiles as well?

It is not evident which cells of the dataset are being employed to perform the CCC analysis. Are the interactions ranked across all pairs of interacting cells?

How are odds ratios computed? A cytokine receptor can appear in several top ranked interactions, how are these “duplicates” treated?

P5. We have now provided additional details in the Methods section for the ‘Cytokine activity agreement’ analysis in accordance with the reviewers’ example questions.

CytoSig is a collection of consensus, data-driven, cytokine-activity signatures compiled using a compendium of transcriptomic profiles ⁷. We used CytoSig’s 43 high-quality signatures to infer which cytokines induce signalling activities in each cell type. We then used this information to assess if a cytokine-receptor interaction reported by the different CCC methods was supported by the corresponding cytokine downstream signalling activities.

We computed the cytokine activity scores for all cell types with the multivariate linear regression model (‘mlm’) method of decoupleR at the pseudobulk level. We chose the mlm method as an approach that models the effect of multiple cytokines and that performed best in a recent footprint-focused analysis benchmark ⁸.

To build the pseudobulk profiles ~~for each cell type~~, we log₂-transformed the summed counts within each cell type, and kept only genes which were expressed in at least 10% of the cells and with a summed raw count above 5.

In this evaluation, we used both the autocrine and paracrine CCC predictions, calculated using expression counts at the cell-type level for all cell types, from the HER2+ and triple negative breast cancer subtype datasets ⁹. We considered

any cytokine signature with a positive score and FDR-corrected p-value ≤ 0.05 in the target cell types as an active cytokine (~~ground truth~~). We considered all CCC predictions with a ligand corresponding to a CytoSig signature, including the same ligand to multiple receptors, matched to any of the aliases of the cytokines. Odds ratios were then calculated as the ratio between any CCC prediction with corresponding active cytokine in a given receiver cell type, and those assigned to the negative class - i.e. the remainder of the cytokine signatures.

As this analysis is treated like a classification problem, it could be of interest providing the results based on ROC/AUC statistics rather than on a Fisher's exact test. This could provide an approximated idea of the trade-off (true positives vs false positives i.e. active/inactive cytokines) yielded by the different CCC methodologies.

P6. As the reviewer points out, the set up for our analyses is indeed a classical classification problem and AUROC/AUPRC curves would have been the go-to assessment statistic. In fact, they were also our first choice but we then noticed some limitations with using them in our agreement analyses. Namely, in our case the different filtering approaches (e.g. SingleCellSignalR vs CellPhoneDB), or the lack of any such filtering (e.g. NATMI), would mean the AUROC curves are built on predictions with large differences in the total number of interactions coming from different methods. Thus, we opted to instead use the Odds ratios; similarly to an AUROC curve, the Odds ratio analysis considers ranked predictions across different thresholds, but focused only on the ratio between true positives and false positives.

That said, we agree that this would not provide a complete picture and, as suggested by the reviewer, we now generated AUROC/AUPRC curves of the 'Agreement with Cytokine activities'.

We first built AUROC/AUPRC curves for each method using their independent universes, meaning that each method's performance was estimated using only the interactions returned by that method.

Albeit informative of the predictive ability of methods, considering the independent universe of interactions for each method means that their performance is not directly comparable. We therefore also generated AUROC/AUPRC curves in which all methods are evaluated using the union of all predictions, and are hence directly comparable. Any missing (filtered) interactions in this case were max-imputed, i.e. assigned the worst predictor value (**Supp. Fig. S14**). Thus, once bound to the same universe the performance

of the scoring functions used by the different methods is largely comparable. Nevertheless, this setting would discredit the added value of false positive filtering. Therefore, we include both figures, as both have specific limitations and complementary content.

We have now included the AUPRC/AUROC estimates (**Supp. Fig. S14**):

Supplementary Figure S14. AUROC/AUPRC of CCC Method Agreement with Cytokine Activities. **A)** Each method's performance was estimated independently, i.e. only using the interactions predicted following the preprocessing steps of each method. **B)** Methods were evaluated using the union of all interactions predictions, and missing interactions were max-imputed. To ameliorate the differences of predictions assigned to the negative class between the different methods, we included a subsampling step, in which we downsampled the negative class 100 times to match the (lower) number of interactions assigned to the positive class.

More globally, in the in vivo datasets, it should be expected that a cell is influenced by multiple cytokines. However, as far as we understand, combined response signatures are not taken into account.

P7. We agree with the reviewer that cytokines affect cells in combinations. In fact, this was one of our main motivations to choose the 'mlm' method, since a multivariate linear regression attempts to model signature expression as a combination of the effect of multiple cytokines ⁷. We acknowledge that this was not clear, and we have now expanded the description in the methods section (see P5).

We recommend the authors to clearly highlight the limitations of using such signatures as ground truth.

P8. We agree with the reviewer that it is important to highlight the limitations. Accordingly, we extended the paragraph highlighting the limitations of our analyses.

Our agreement analyses are based on assumptions that are only approximations of reality. The limitations include the restricted coverage of the cytokine activity signatures and receptor proteins, the technical shortcomings of current spatial transcriptomics technologies. Furthermore, such benchmarks cannot distinguish simple co-expression from actual CCC events, and do not capture complex relationships between CCC events. Since a gold standard is currently not available and the biological ground truth is largely unknown ^{10,11}, our analyses cannot give a definitive answer of what method is best. However, we believe that these results are useful to indirectly ~~evaluate the methods' predictions.~~ support the methods' predictive potential.

We also remove the statement in the results about using the comparison to other modalities as a means to assess the methods' accuracy:

~~The results of the previous section showed low overlap in the results obtained by using different methods and resources, which raises the question of which of them are more accurate.~~ Next, given the lack of a ground truth, we used other data modalities to indirectly evaluate the methods using OmniPath, the resource with the largest coverage.

- “Comparison of predictions with other data modalities” using spatial data. We find a conceptual misunderstanding in the use of spatial colocalisation to benchmark CCC methods. The “assumption that their highest ranked interactions should be positively associated in interactions between pairs of adjacent cell types” is misleading.

P9. We agree with the reviewer that the phrasing of our assumption in the spatial benchmark was incorrect and as stood implied a conceptual misunderstanding.

What we wanted to say is: rather than a direct relationship between interaction rank and spatial adjacency, a certain proportion of the expression patterns, and hence co-expression between interacting cell populations, is reflected by spatial information. This is based on the assumption that, while many other factors are involved, co-localized cell populations have a higher chance to interact with each other than other non-adjacent cell types^{1,12-14}.

To this end, we have now changed the text describing our assumption:

Next, we leveraged spatial information as a way to support ~~benchmark~~ the methods’ predictive potential, under the assumption that, while many other factors are involved, co-localized cell populations are expected to have a higher chance to interact with each other than other non-adjacent cell types^{1,12-14}. That is, the highest ranked interactions predicted between various cell populations are expected to be positively associated in interactions between pairs of adjacent cell types (Methods 4.6.2).

Molecules involved in cell-cell interactions usually do not have a 1:1 specificity, however multiple combinations of L/R or interacting adhesion molecules exist. Just to mention a few examples, collagens/integrins or TGFb/Notch/WNT signalling molecules may work through different combinations. Let’s consider the following scenario:

(i) Interactions the resource: JAG1-NOTCH1, JAG1-NOTCH2, JAG2-NOTCH1, JAG2-NOTCH2. Note that there are more Notch receptors (e.g. NOTCH3), but let’s consider only these for simplicity.

(ii) Transcriptomic profile: cell1 overexpressing JAG1+, cell2 overexpressing NOTCH1+, cell3 overexpressing JAG2+, cell4 overexpressing NOTCH2+. Assume overexpression occurs at similar levels.

(iii) Colocalization: cell1 and cell2 colocalise, but not cell3 or cell4.

Under these conditions, one can not expect a CCC method to rank higher cell1-cell2 over cell1-cell4 or cell3-cell2 or cell3-cell4 without prior knowledge on neighbouring cells.

P10. We thank the reviewer for sharing this thoughtful example, that we visualised below.

Response Figure 3. Schematic of spatial analysis limitation example.

We agree with the reviewer that in reality CCC events are not simply 1:1 interactions and that the scenario depicted above would likely lead to incorrect C1-C4 and C3-C2 predictions. We see this as a general limitation of inferring cell-communication from dissociated single-cell data.

As the reviewer points out, our spatial agreement analysis does not necessarily distinguish the specificity of interactions. As we mentioned above, it is just expected to partially reflect the general coordination of adjacent cell types, which is supported by the observed results.

We acknowledge that as suggested by the reviewer's example, our spatial analysis has a number of limitations and we certainly agree that it does not represent a gold standard for the evaluation of methods. For this reason, we generally refrained from over-interpreting the results of the agreements with other modalities as comparison between the methods. Instead, in our discussion we used the agreement with the other modalities to highlight the ability of the methods to detect biologically-relevant signals, and hence their value as tools to

prioritise the CCC predictions relevant for subsequent validation. We have now underlined this, as pointed out in P8.

Furthermore, we now refrain from using ‘ground truth’ in regards to the ‘assumed truth’ and our agreement analyses, respectively.

Accordingly, we do not consider that spatial data represents a gold standard dataset for comparing CCC methods. Instead, it can be very valuable to refine CCC predictions. Instead, information on pairs of adjacent cell types can be used to discard interactions that are infeasible due to physical distance between the cells.

We fully agree with the reviewer that spatial information should rather be used to constrict or discard ‘erroneous interactions’ (visualised above), and we have explicitly noted this in the discussion:

[...] similarly to previous efforts, we used spatial information to support the methods’ predictions^{1,12}. We saw that most methods prioritise interactions between colocalized cell types, and this was much clearer in the well-structured brain cortex than in breast cancer tissue. These results suggest that the performance of the methods depends on the type of tissue, and that, if available, spatial information should be used to inform^{15,16} or constrain¹⁷ the predictions.

- The low performance of CellChat, CellPhoneDB and SingleCellSignaR, in at least one of the datasets employed, might be due to the fact that the composite scores employed by these tools do not allow for a direct comparison between scores generated for different ligand-receptor combinations. The scores of these tools depend on the input value of the ligands and receptors, while other methods rely on scaled scoring methodologies allowing for a more direct comparison between pairs of ligands and receptors. Can the authors elaborate on this?

PII. The reviewer raises a fair and relevant point, that we agree we should further elaborate on.

A main conceptual difference between CellChat, CellPhoneDB and SingleCellSignaR and the rest of the methods is that the aforementioned tools provide clear thresholds to account for false positives. Hence, as the reviewer points out, ranking may indeed not be the main focus of their scoring functions.

Thus, one could consider that using this false-positive filtering implies that all interactions that are not filtered should be considered equally relevant. However, the number of unfiltered interactions is often too large to evaluate or interpret on its own. Thus, following the recommended filtering for each of these methods, we opted to then rank their scores as a way to distinguish between the large number of predictions.

Saying this, we certainly agree with the reviewer that we missed an opportunity to highlight the value of the false positive thresholds in the case of CellChat and CellPhoneDB. We have now highlighted this in our revised text:

Connectome, the Crosstalk scores, and NATMI showed a consistent trend across both datasets, while the consensus of the methods, logFC Mean, CellChat, CellPhoneDB, and SingleCellSignalR (**Table 1**) showed negative or lack of signal for the higher ranks of the HER2+ dataset (**Figure 6A**). Notably, a high agreement with Cytokine activities was observed for CellChat and CellPhoneDB in the HER2+ dataset, when considering all of their predictions subsequent to false-positive filtering (vertical line in **Figure 6A**), highlighting the value of the false-positive control steps of these methods.

We also underline the added value of the false positive filtering in Table 1:

“# CellPhoneDB, CellChat, and SingleCellSignalR provide explicit thresholds to control for false positive interaction predictions. In the case of CellPhoneDB, and CellChat, permutation-based p-values are used to control the false-positive rate, whereas SingleCellSignalR’s has a suggested threshold of LRscore \geq 0.5.”

- Some result sections would benefit from highlighting their biological implications. For example “the similarity among the resources was generally higher when considering transmitters and receivers, rather than the interaction themselves” one could add, for example, “meaning that different resources account for different interactions for the same proteins”.

P12. We agree with the reviewer that our manuscript would benefit from extending the biological implications of our results, and we now have added the reviewer’s suggested example, among others, to our text:

Page 8:

“...the similarity among the resources was generally higher when considering transmitters and receivers (Supp. Figure S1-2), rather than the interaction themselves, suggesting that different resources account for different interactions between the same proteins.”

Page 10:

In summary, our results indicated biases towards certain pathways, functional categories, and tissue markers across the different CCC resources, implying that resource choice can influence the functional interpretation of CCC predictions.

Page 15:

These results suggest that the interactions identified as relevant by all methods were largely concordant with cytokine activities, confirming the agreement of predicted CCC interactions with downstream signalling events.

Page 16 (now 17):

In summary, our results showed a positive association of interactions predicted by most methods and spatially-adjacent cell types in the well-structured brain cortex, while the associations were less consistent in the breast cancer subtypes. This positive association suggests that, despite the dissociation of single-cells and their grouping into cell types, CCC predictions partly reflect the expression patterns encoded by tissue spatial context.

- In page 11, what is a “functional cell state”? How can a “cell state” be underrepresented in a CCC database?

P13. We thank the reviewer for their comment; we realise that using cell state is confusing in particular in the context of single-cell data. The authors of CancerSEA¹⁸ called “functional cell state” a collection of curated, consensus gene-sets characterising cancer functional states. To avoid confusions, we don’t use this term and simply refer them as cancer-related gene sets:

Finally, we matched interactions to cancer-related gene sets from CancerSEA¹⁸, which were also observed to be unevenly represented.

Bibliography

1. Jin, S. *et al.* Inference and analysis of cell-cell communication using CellChat. *Nat. Commun.* **12**, 1088 (2021).
2. Türei, D. *et al.* Integrated intra- and intercellular signaling knowledge for multicellular omics analysis. *Mol. Syst. Biol.* **17**, e9923 (2021).
3. Noël, F. *et al.* Dissection of intercellular communication using the transcriptome-based framework ICELLNET. *Nat. Commun.* **12**, 1089 (2021).
4. Efremova, M., Vento-Tormo, M., Teichmann, S. A. & Vento-Tormo, R. CellPhoneDB: inferring cell-cell communication from combined expression of multi-subunit ligand-receptor complexes. *Nat. Protoc.* **15**, 1484–1506 (2020).
5. Baccin, C. *et al.* Combined single-cell and spatial transcriptomics reveal the molecular, cellular and spatial bone marrow niche organization. *Nat. Cell Biol.* **22**, 38–48 (2020).
6. Zhang, Y. *et al.* Cellinker: a platform of ligand-receptor interactions for intercellular communication analysis. *Bioinformatics* (2021) doi:10.1093/bioinformatics/btab036.
7. Jiang, P. *et al.* Systematic investigation of cytokine signaling activity at the tissue and single-cell levels. *Nat. Methods* **18**, 1181–1191 (2021).
8. Badia-i-Mompel, P. *et al.* decoupleR: Ensemble of computational methods to infer biological activities from omics data. *BioRxiv* (2021) doi:10.1101/2021.11.04.467271.
9. Wu, S. Z. *et al.* A single-cell and spatially resolved atlas of human breast cancers. *Nature Genetics* (2021).
10. Armingol, E., Officer, A., Harismendy, O. & Lewis, N. E. Deciphering cell-cell interactions and communication from gene expression. *Nat. Rev. Genet.* **22**,

- 71–88 (2021).
11. Almet, A. A., Cang, Z., Jin, S. & Nie, Q. The landscape of cell-cell communication through single-cell transcriptomics. *Current Opinion in Systems Biology* **26**, 12–23 (2021).
 12. Hu, Y., Peng, T., Gao, L. & Tan, K. CytoTalk: De novo construction of signal transduction networks using single-cell transcriptomic data. *Sci. Adv.* **7**, (2021).
 13. Armingol, E. *et al.* Inferring the spatial code of cell-cell interactions and communication across a whole animal body. *BioRxiv* (2020) doi:10.1101/2020.11.22.392217.
 14. Palla, G., Fischer, D. S., Regev, A. & Theis, F. J. Spatial components of molecular tissue biology. *Nat. Biotechnol.* (2022) doi:10.1038/s41587-021-01182-1.
 15. Fischer, D. S., Schaar, A. C. & Theis, F. J. Learning cell communication from spatial graphs of cells. *BioRxiv* (2021) doi:10.1101/2021.07.11.451750.
 16. Tanevski, J., Ramirez Flores, R. O., Gabor, A., Schapiro, D. & Saez-Rodriguez, J. Explainable multi-view framework for dissecting inter-cellular signaling from highly multiplexed spatial data. *BioRxiv* (2020) doi:10.1101/2020.05.08.084145.
 17. Garcia-Alonso, L. *et al.* Mapping the temporal and spatial dynamics of the human endometrium in vivo and in vitro. *Nat. Genet.* **53**, 1698–1711 (2021).
 18. Yuan, H. *et al.* CancerSEA: a cancer single-cell state atlas. *Nucleic Acids Res.* **47**, (2018).

REVIEWERS' COMMENTS

Reviewer #1 (Remarks to the Author):

Happy with the revision. No more comments.

Reviewer #3 (Remarks to the Author):

The authors have addressed all of the technical concerns raised in the second round of review.

REVIEWERS' COMMENTS

Reviewer #1 (Remarks to the Author):

Happy with the revision. No more comments.

Response: We thank the reviewer for their positive and insightful comments and their added value to our work.

Reviewer #3 (Remarks to the Author):

The authors have addressed all of the technical concerns raised in the second round of review.

Response: We thank the reviewer for the insightful and in-depth review that helped us markedly improve our manuscript.